

# Optimization of weather forecasting for cloud cover over the European domain using the meteorological component of the Ensemble for Stochastic Integration of Atmospheric Simulations version 1.0

Yen-Sen Lu[1], Garrett Good[2], and Hendrik Elbern[3]

[1]Institute of Energy and Climate Research – Troposphere (IEK-8), Forschungszentrum Jülich GmbH, 52425 Jülich, Germany
[2]Fraunhofer Institute for Energy Economics and Energy System Technology IEE, Königstor 59, 34119 Kassel, Germany
[3]Rhenish Institute for Environmental Research at the University of Cologne, Cologne, Germany

**Correspondence:** Yen-Sen Lu (ye.lu@fz-juelich.de)

**Abstract.**

In this study, we present an expansive sensitivity analysis of physics configurations for cloud cover using the Weather Forecasting and Research Model (WRF V3.7.1) on the European domain. The experiments utilize the meteorological part of a large ensemble framework known as the Ensemble for Stochastic Integration of Atmospheric Simulations (ESIAS-met). The experiments first seek the best deterministic WRF physics configuration by simulating over 1,000 combinations of microphysics, cumulus parameterization, planetary boundary layer physics (PBL), surface layer physics, radiation scheme and land surface models. The results on six different test days are compared to CMSAF satellite images from EUMETSAT. We then selectively conduct stochastic simulations to assess the best choice for ensemble forecasts. The results indicate a high variability in terms of physics and parameterization. The combination of Goddard, WSM6, or CAM5.1 microphysics with MYNN3 or ACM2 PBL exhibited the best performance in Europe. For probabilistic simulations, the combination of WSM6 and SBU–YL microphysics with MYNN2 and MYNN3 showed the best performance, capturing the cloud fraction and its percentiles with 32 ensemble members. This work also demonstrates the capability and performance of ESIAS-met for large ensemble simulations and sensitivity analysis.

## 1 Introduction

Recent events in 2021 have highlighted the destructive potential of extreme weather to human life and property. The prevention of such loss has demonstrated the urgent need for better weather forecasting and research to understand these weather conditions, especially with respect to how they are caused by climate change (Tabari, 2020; Palmer and Hardaker, 2011; Bauer et al., 2015; Sillmann et al., 2017; Samaniego et al., 2018; Bellprat et al., 2019; Bauer et al., 2021). In the energy sector, better





weather predictions can help prevent damage to power infrastructure, but also to combat climate change in the first place by
facilitating the integration of higher proportions of weather-dependent renewables into the power system (e.g. Rohrig et al.,
2019; Adeh et al., 2019), for which unexpected weather can be incredibly expensive for grid operators and result in, for exam-
ple, negative wind energy prices. Moreover, better forecasting is also needed to study to the impact of energy on ecology Yan
et al. (2018); Lu et al. (2021).

There are generally two types of weather predictions: deterministic and probabilistic simulations (Palmer, 2012). Determin-
istic simulations rely on the accuracy of a single simulation to capture the values of meteorological variables. Improving the
accuracy of such weather and climate models involves research to improve numerical solvers, accurate parameters, and ad-
vanced governing equations. Various global and regional weather models are developed by national and international weather
agencies. However, the optimal implementation of any model can vary greatly for different regions. The widely used Weather
Research and Forecast (WRF) model, a publically available research software system, is, for example, developed in North
America, where the optimal meteorological models, or even the optimal parameterization of land types given (e.g. the typical
density and size of structures), can differ in Europe and elsewhere.

Sensitivity analysis is a widely accepted method for identifying the most suitable model composition. The improvement
of high-performance computation has enhanced the ability to perform higher resolution simulations, or larger simulations for
sensitivity analysis (Borge et al., 2008; Jin et al., 2010; Santos-Alamillos et al., 2013; García-Díez et al., 2013; Mooney et al.,
2013; Warrach-Sagi et al., 2013; Kleczek et al., 2014; Pieri et al., 2015; Stergiou et al., 2017; Gbode et al., 2019; Tomaszewski
and Lundquist, 2020; Varga and Breuer, 2020). To date, most sensitivity analyses are based on a small number of combinations
of physics configurations. The largest sensitivity analysis of WRF, for example, includes 63 physics combinations (Stergiou
et al., 2017), whereas WRF has over 1 million possible combinations of 23 microphysics, 14 cumulus settings, 13 planetary
boudary layer (PBL) physics, 7 land surface and 8 surface layer models, 8 longwave and 8 shortwave radiation schemes
(Skamarock et al., 2008). However, it is not enough to optimize WRF by sensitivity analysis on a few configuration sets. Most
physics combinations can be expected to be biased to the measurement from ground observations, making them unsuitable for
deterministic forecasts that should perform well on average.

Probabilistic simulations are meant to address the spread of possible solutions, accounting for uncertainty from initial condi-
tions or modelling physics by performing large multiphysics ensemble simulations (e.g. Li et al. (2019)) or employing stochas-
tic schemes in light of greater high-performance computing power (Ehrendorfer, 1997; Palmer, 2000; Dai et al., 2001; Gneiting
and Raftery, 2005; Leutbecher and Palmer, 2008; Hamill et al., 2013). While deterministic forecasts should provide the most
likely case, probabilistic forecasts are likely to capture the uncertainty of that solution, including rare weather extremes. The
challenge of ensemble simulation is thus not only the scientific challenge (e.g. proper scoring rules, Sillmann et al., 2017) but
also the technical challenge, for example large supercomputing facilities that can handle simulations large enough to capture
the most extreme and damaging events typically missed by contemporary, $O(10)$ member simulations.

The urgent need for better probabilistic simulations is the motivation behind the development of the Ensemble for Stochas-
tic Integration of Atmospheric Simulations (ESIAS) (Berndt, 2018; Franke et al., 2022). We have developed ESIAS to carry
out ultra-large ensemble forecasts of $(O)1000$ members and have further integrated stochastic schemes to generate simula-





tion members for data assimilation, both for probabilistic simulations with stochastic schemes and for sensitivity analysis. Moreover, ESIAS aims to cope with future *exascale* computation requirements in order to perform forecasts that are not yet operationally possible. We use ESIAS to perform a sensitivity analysis of over 1,000 WRF physics combinations in Europe, with a focus on wind and solar energy and to provide recommendations for future weather research.

The object of this study is to optimize ESIAS-met by determining the most suitable physics configuration to better perform
the simulation of the cloud cover by comparing simulation results to the satellite measurements. We also perform simulations combining multi-physics simulation and stochastic simulation to achieve the probabilistic simulation to capture the cloud cover condition over the European domain.

In this article, we introduce the ensemble weather forecasting system ESIAS and its meteorological component in section 2. This section describes the methods applied for the sensitivity analysis regarding the model physics configuration and the
description of the methods for evaluating the simulation results. Section 3 describes the data used in this study. The outcome of the three sets of sensitivity analyses is shown in section 4 and the results are discussed and concluded in the section 5.

## 2   Model description

### 2.1   Modeling system: ESIAS-met v1.0

ESIAS is the stochastic simulation platform developed by IEK-8 at Forschungszentrum Jülich and by the Rhenisch Institute for
Environmental Research at the University of Cologne. ESIAS consists of two parts, which are based on the Weather Research and Forecasting (WRF) model V3.7.1 (Skamarock et al., 2008) and the EURopean Air pollution Dispersion  Inverse Model (EURAD–IM, Franke et al. (2022)), which we shall refer to as ESIAS-met and EISAS-chem, respectively. The full details of these two models are described by Berndt (2018) and Franke (2018).

Figure 1 illustrates the full concept of ESIAS-met, based on WRF V3.7.1. The ESIAS System Control Scripts are used
to control the WRF Preprocessing System to produce the intermediate meteorological inputs to generate WRF inputs and boundary data for the simulation. The namelist of ESIAS-met is the same as WRF V3.7.1 and thus the input and output filenames are flexible for different ensemble simulation strategies. The ESIAS System Control Scripts includes a namelist generator based on the ecosystem of ESIAS-met to better fit the system.

The ESIAS-met executables apply the Message Passing Interface (MPI) to perform large ensemble simulations and are
thus advantageous for large ensemble simulations with interactive members on HPCs. (Large individual ensemble simulations on the HPCs is inhibitive and will cause long queuing times). The main purpose of ESIAS-met is to perform large ensemble simulations based on stochastic schemes. The stochastically perturbed parameterization tendency (SPPT) scheme (Buizza et al., 1999) and the stochastic kinetic energy backscatter scheme (SKEBS) (Berner et al., 2009, 2011) are therefore used.





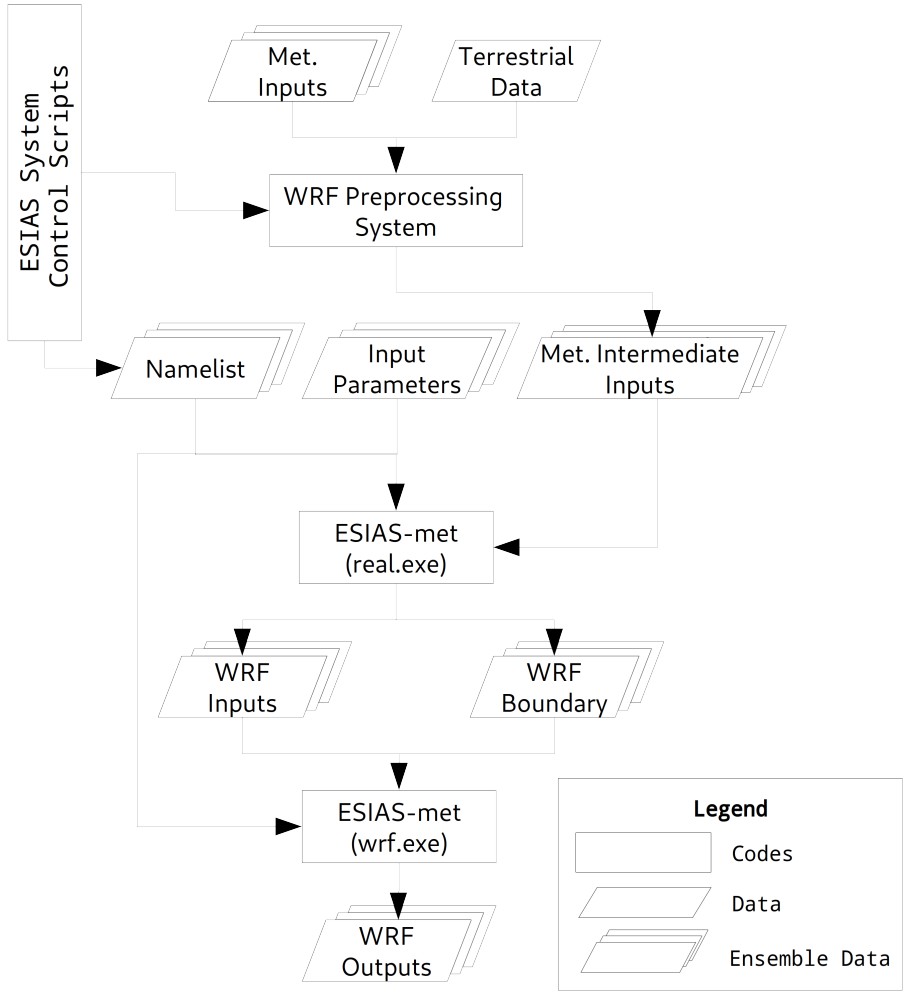

**Figure 1.** The concept and entire process of the ESIAS-met scheme. The ESIAS System Control Scripts control the WRF Preprocessing System and the namelist to generate necessary inputs.

## 2.2 Model setup

ESIAS-met is run with boundary conditions from the Global Ensemble Forecasting System (GEFS) and the MODIS land use data for generating meteorological intermediate inputs. The map projection is Lambert Confromal with a central point of $54^o N, 8.5^o W$. The horizontal resolution is 20 km, and the number of horizontal gridpoints are 180 by 180. The vertical layers consist of 50 grid points, which are unevenly spaced for the first 11 layers near the boundary layer. We do not use the nested domain to generate finer grids in higher resolution due to the high computational demand. A previous study and large sensitivity

analysis by Stergiou et al. (2017), which tested 68 cases of different physics configurations, performed the simulation on the European domain but with a different approach vis-á-vis the WRF physics configuration.





We generate a very large ensemble simulation using different physics schemes to investigate the optimal physics configuration for the simulation output. We generate three sets of large ensemble simulations based on different combinations of physics schemes. The first set (Set 1) investigates combinations of microphysics, the cumulus parameterizations, and planetary boundary layer physics. The physics configurations used are listed in Table 1. It is recommended that the surface layer physics be set with planetary boundary layer physics in WRF. The five-layer thermal diffusion for land surface physics, Dudhia shortwave radiation physics, and RRTM longwave radiation physics are employed in this set of numerical experiments. Set 1 comprises a total of 672 ensemble simulations for the optimization.

The second set (Set 2) adds combinations of land surface models, shortwave radiation schemes, and longwave radiation schemes to a combination of microphysics, cumulus parameterizations, and planetary boundary layer physics selected based on the results from Set 1 (Table 2). In Set 2 there are 513 ensemble simulations in total. For the acronyms of WRF physics and paratermizations, view Table A1.

The PBL ACM2 only considers the surface layer physics of MM5 similarity and therefore does not apply to the other surface layer physics models. The combination of Eta similarity and the CLM4 land surface model is ruled out as it does not employ surface layer physics Eta similarity. For the longwave and shortwave radiation physics, we have only three combinations of physics configurations: RRTM and Dudhia, RRTMG and RRTMG, and Goddard and Goddard for shortwave and longwave radiation physics scheme, respectively. For surface layer physics, we employ Monin–Obukhov similarity and therefore the revised MM5 similarity and Eta similarity schemes. The MYNN surface layer scheme is used to investigate its suitability with the MYNN2 and MYNN3 PBL physics. The sophisticated land surface models CLM (version 4) and Noah LSM are tested as well as RUC LSM, which performs similarly to the other two LSM (Jin et al., 2010).

**Table 1.** Employed physics configuration of Set 1, which summarized $12 \times 7 \times 8$ configurations. The full description of abbreviations can be found in Table A1.

| Microphysics | Cumulus Parameterization | PBL & | Surface layer physics |
|---|---|---|---|
| Kessler | Kain-Fritsch | YSU | MM5 similarity |
| Lin (Purdue) | Betts-Miller-Janjic | MYJ | Eta similarity |
| WSM3 | Grell-Freitas | GFS | Pleim-Xiu |
| WSM5 | Simplied Arakawa-Schubert | QNSE | QNSE surface layer |
| Eta (Ferrier) | Grell-3 | MYNN2 | MM5 similarity |
| WSM6 | Tiedtke | MYNN3 | MYNN surface layer |
| Goddard | New SAS | ACM2 | MM5 similarity |
| Thompson | | BouLac | MM5 similarity |
| Milbrandt 2-mom | | | |
| Morrison 2-mom | | | |
| CAM 5.1 | | | |
| SBU–YLin | | | |





**Table 2.** Employed physics configuration of Set 2,which summarized $3 \times 3 \times 3 \times 19$ configurations. The full description of abbreviations can be found in Table A1 and Table A2.

| Microphysics | Cumulus | PBL | Radiation(SW&LW) | Surface Layer | LSM |
|---|---|---|---|---|---|
| WSM5 | Kain-Fritsch | MYNN2 | RRTM+Dudhia | MM5 | Noah |
| WSM6 | Grell-3D | MYNN3 | RRTMG | Eta similarity | RUC |
| Goddard | Tiedtke | ACM2 | New Goddard | MYNN | CLM4 |

The third set (Set 3) is used to understand the impact of the physics configurations on the ensemble simulations. We also perform a small number of ensemble simulation with a size of $O(32)$ by SKEBS for each physical configuration, which is based on the combination of different microphysics and PBL physics listed in Table 3. According to Jankov et al. (2017) and Li et al. (2019), SKEBS can produce a large ensemble spread and we only use it to identify the extent of the spread that one single stochastic scheme can produce. One ensemble member of the 32 is not perturbed in order to serve as a control run. All Set 3 simulations employ the Grell-3 cumulus parameterization, the Dudhia shortwave radiation physics, the RRTM longwave radiation physics and the RUC land surface model for land surface physics.

**Table 3.** Employed physics configuration of set 3

| Microphysics | PBL & | Surface layer physics |
|---|---|---|
| Kessler | YSU | MM5 similarity |
| WSM3 | GFS | Pleim-Xiu |
| Eta (Ferrier) | MYNN2 | MM5 similarity |
| WSM6 | MYNN3 | MYNN surface layer |
| Goddard | ACM2 | MM5 similarity |
| Thompson | BouLac | MM5 similarity |
| Morrison 2-mom | | |
| CAM 5.1 | | |
| SBU–YLin | | |

We use the term *cluster* to refer to a set of physics configurations sharing a specific option, for instance all configurations sharing a particular microphysics scheme.

## 2.3 Model performance evaluation and measurements

### 2.3.1 Ternary determination of cloud mask

To determine the accuracy of the cloud cover prediction, we apply the determination method and separate the cloud cover fraction into three gradation of grid conditions: clear-sky ( $< 5\%$ ), partially cloudy ( $\geq 5\%$ and $< 95\%$ ), and fully cloudy ( $\geq 95\%$ ) to show more details beyond clear sky and cloudy. The definition of clear sky follows the ASOS definition (Diaz





et al., 2014) as 5% cloud fraction, while full cover is defined as 95% cloud fraction. The gradation of partial clouds can show more details for both simulation and satellite data, which we then compare. Table 4 shows the detection rate of cloud cover conditions and the deterministic result from the ternary detection. The traditional binary detection classifies the outcome into just three categories: false (overpredict), miss, and match. Our determination increases this to five categories to capture more detail in the partially cloudy areas, as well as the prediction ability for the different physics configurations.

**Table 4.** The five possible outcomes for the detection of predicted cloud cover determined by the observed cloud cover, where 0% - 5% , 5% - 95% , 95% - 100% are defined as clear, partial, and full cover, respectively. Indaddition to the classifications as match, miss, false/over-predict in the detection method for binary condition, we add "over" between match and over-predict and "less" between miss and match for the ternary condition.

|  |  | Prediction | | |
| --- | --- | --- | --- | --- |
|  |  | **clear** | **partial** | **full cover** |
|  | **clear** | Match | Over | Over-predict |
| observed | **partial** | Less | Match | Over |
|  | **full cover** | Miss | Less | Match |

### 2.3.2 Kappa score

The Kappa ($\kappa$) score is first used to determine the agreement between two or more raters, which is used to rate the score of an object by determination, in large data sets such as from subjects in psychological research (Fleiss and Cohen, 1973). This score for deterministic results is widely used in natural sciences such as land science for determining the change of land use (e.g. Schneider and Gil Pontius (2001), Yuan et al. (2005), and Liu et al. (2017) ) or machine learning for scoring and validation
(e.g. Dixon and Candade (2008) and Islam et al. (2018)). The equation of the Kappa score for multiple raters is calculated as:

$$\kappa = \frac{\bar{P} - \bar{P}_e}{1 - \bar{P}_e} \tag{1}$$

where $\bar{P}$ is the sum of $P_i$, the matching rate of the $i^{th}$ subjects for $k$ categories and $\bar{P}_e$ is the sum of the category rate $p_j$ over $j$.

$$\bar{P} = \sum_{i=1}^{N} \frac{1}{n(n-1)} [\sum_{j=1}^{k} n_{ij}^2 - (n)] \tag{2}$$

$$p_j = \frac{1}{n(n-1)} [\sum_{j=1}^{k} n_{ij}^2 - (n)] \tag{3}$$





### 2.3.3 Kernel density estimation

The kernel density estimation (KDE) is a method to approximate the probability density function of data. A variable $X$ with $n$ independent data points $x_1$, $x_2$, ..., $x_n$ at $x$ can be expressed as

$$f_h(x) = \frac{1}{nh} \sum_{i=1}^{n} K(\frac{x_i - x}{h}) \tag{4}$$

, where $h$ and $K$ are the bandwidth and kernel functions, respectively. The Kernel function $K(u)$ can be uniform $\frac{1}{2}I(|u| \leq 1)$ or Gaussian $\frac{1}{\sqrt{2\pi}} \exp(-\frac{1}{2}u^2)$ for different proposes. In our study we use the Gaussian kernel. Here we also propose normalizing the KDE with the cumulative KDE with $x$ in the range from 1 to $m$ as:

$$f_{h,acc}(i) = \sum_{j<i} f_h(x_j), \text{for } i = 1, 2, ..., m \tag{5}$$

The resulting cumulative KDE can be normalized by the last item of $f_{h,acc}(i)$ i.e. $f_{h,acc}(x_m)$, and therefore a normalized
cumulative KDE can be used to show the cumulative probability distribution of the data, which increases monotonically.

## 3 Data description

### 3.1 Input data

The initial and boundary conditions are generated from the control data of the Global Ensemble Forecast System (GEFS) of the National Centers for Environmental Prediction (NCEP) (Hamill et al., 2013). This dataset has approximately 40 km resolution
and 42 vertical levels. The detail on the GEFS data is described by Hamill et al. (2011). To better represent forecasting skill from April to September, we simulate 2015-04-13, 2015-05-15, 2015-06-17, 2015-07-19, 2015-08-23, and 2015-09-21. For each simulation we use two days with 3-hourly forecasting fields based on the reforecasting data from GEFS. The soil texture and land use condition are based on the 2-minute resolution data and Moderate Resolution Imaging Spectroradiometer (MODIS) 30-second resolution for Noah-modified 20-category IGBP-MODIS land use, respectively. The target domain covers
most of Europe, and is a single domain with a 20 km horizontal resolution. Figure 2 shows the whole area of the target domain and the elevation height.

### 3.2 Satellite data

To validate and rate the model performance, we use the Cloud Fraction Cover (CFC) product from the EUMETSAT Climate Monitoring Satellite Application Facility (CMSAF) (Stengel et al., 2014). This product covers the spatial domain of Europe and
Africa and also dates of simulations in 2015, although this product was discontinued after 2018-03-05. The data are cropped to the European domain. Since the CFC data do not include the northern part of Europe, we do not use the whole setup domain for this analysis.

The horizontal resolution (both $0.05^{\circ}$ in longitude and $0.05^{\circ}$ in latitude) of CFC data is higher than the simulation setup of 20km by 20km, which is equivalent to $0.31^{\circ}$ in longitude and $0.18^{\circ}$ in latitude at the center of the model domain. For the

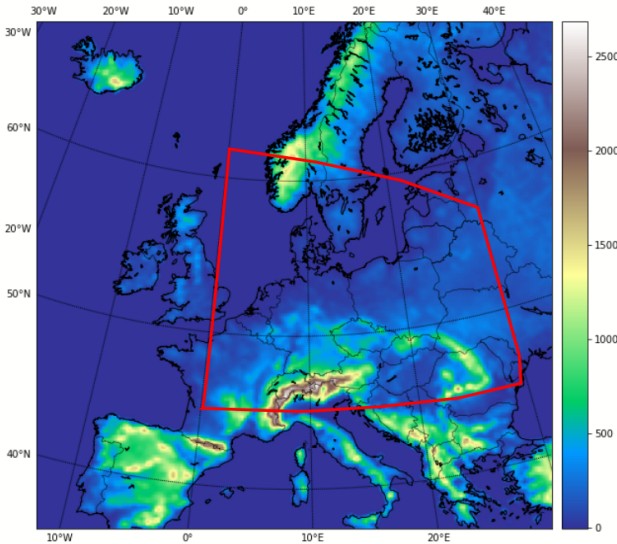

**Figure 2.** The topography of the target domain, Europe domain, for simulation. The red line indicates the area for validating the result with cloud cover from the satellite

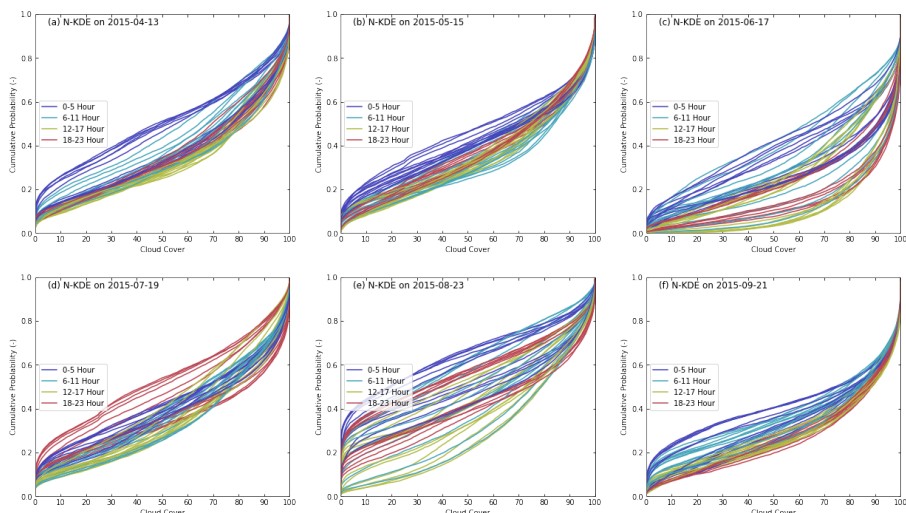

**Figure 3.** The cumulative KDE of cloud cover for the six simulation cases. The colors distinguish the different cloud cover. The different color represent different time period in a day, as shown in the legend.

overall model domain each model grids between 12 and a maximum of 36 grid points from CFC to calculate the average cloud cover fraction for comparison where 99.5% of model grids contains more than 12 grid points from CFC data.

   Figure 3 shows the cloud cover condition of all simulation cases for 48 hours in 2015 within the simulation domain. The blue and orange curves represent the cloud cover during daytime, and the red and cyan curves represent the cloud cover at night.





For the cumulative plots of KDE, which is normalized to 1.0, curves with a higher accumulative rate (rapid growths in y-axis)
represent high clear sky rates in the data, and the curves with a lower accumulative rate (y-axis) depict high full cloud cover. In
the six simulation cases, the cases 2015-06-17 and 2015-08-23 exhibit a very high variation during the 48 hours of simulation
time. Moreover, 2015-06-17 shows a higher cloud cover than 2015-08-23. Cases 2015-04-13, 2015-05-15, 2015-07-19 and
2015-09-21 are similar in the cloud cover condition, since the cumulative distributions show that the variation of cloud cover
is smaller than cases 2015-06-17 and 2015-08-23. However, both case 2015-04-13 and case 2015-07-19 show a higher cloud
cover in the early morning and in the early evening than cases 2015-05-15 and 2015-09-21, respectively. In general, cases
2015-08-23 and 2015-09-21 can represent the cloud cover condition with high variability and low variability, respectively.

## 4   Results

### 4.1   Simulation efficiency

We perform the simulations on JUWELS (Jülich Supercomputing Centre, 2019), the high–performance computer which utilizes
Intel Xeon 24-core Skylake CPUs (48 cores per node) and 96 GiB of main memory. We use 12 CPUs per ensemble member,
8,064 total CPUs for 672 ensemble members and 9,216 CPUs for 768 ensemble members to perform the large ensemble
simulation for each of the sensitivity analyses. These large ensemble simulations are successfully conducted by JUWELS
without performing farming on the HPC, while the stability of large ensemble simulations is guaranteed by ESIAS.

Different physics configurations not only affect the resulting weather fields and states, but also significantly impact the
computing time. Figure 4 and Figure 5 show the average time consumption (solid line) and the range of the time consumption
(color fill) for the configurations of microphysics and planetary boundary layer physics, respectively. We use the simulation case
in 2015-09-21 as the example case only. The most time-consuming simulation is always that with the one CAM5.1 (average
of 26,095 seconds) microphysics and the quickest is with Kessler (average of 10,404 seconds), which only parameterizes
the autoconvection, precipitation cloud, the evaporation of precipitation, and the condensation-evaporation function in the
continuity equation. For the simulation of planetary boundary layer physics, the slowest configuration is QNSE (average of
16,967 seconds) and the fastest is GFS (average of 12,440 seconds). The cumulus parameterization has the smallest effect on
time consumption, with the most time-consuming being Grell-3 (average of 14,715 seconds) and the least time-consuming
being Betts-Miller-Janjic (average 13,447 seconds), where the difference is only 9.4%.

The differences between the first and third quartiles show how much the different physics configurations affect the simu-
lation speed, as shown in Figure 4 (c) and Figure 5 (c). For the physics clusters of microphysics, PBL physics, and cumulus
parameterization, the average quartile difference is 1644.45 seconds, 3,281.25 seconds, and 3,930.54 seconds, respectively.
The outliners are from the CAM5.1 microphysics, which is the most computationally expensive microphysics. The most time-
consuming consideration is from the configuration of microphysics. The average time consumption is similar for the cluster of
PBL physics and the cluster of cumulus parameterization, except for the ONSE PBL physics which consumes 5,000 seconds
more simulation time than other PBL physics. The boxplot confirms that CAM5.1 microphysics and ONSE PBL physics are
most time-consuming physics.





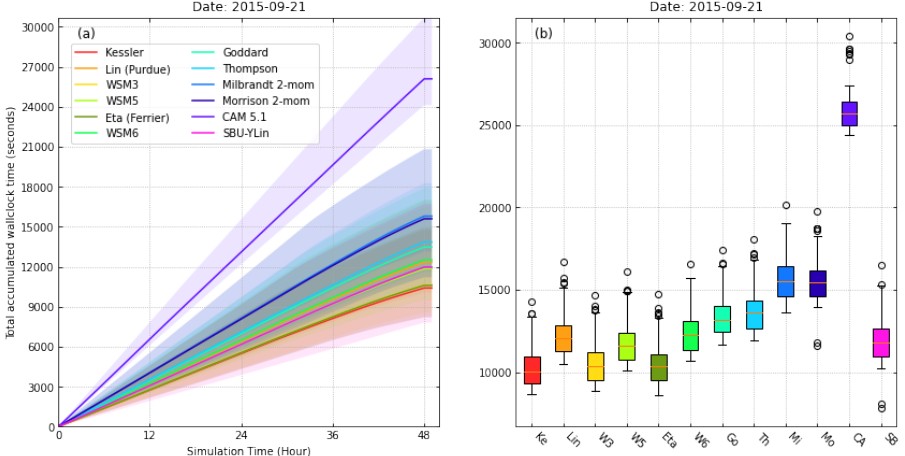

**Figure 4.** The (a) hourly simulation time, (b) total accumulated simulation time, and (c) boxplot by different microphysics configuration within each simulation hour on 2015-09-21. The upper and lower boundaries of the color fill indicates the maximum and minimum simulation time by other physics configurations.

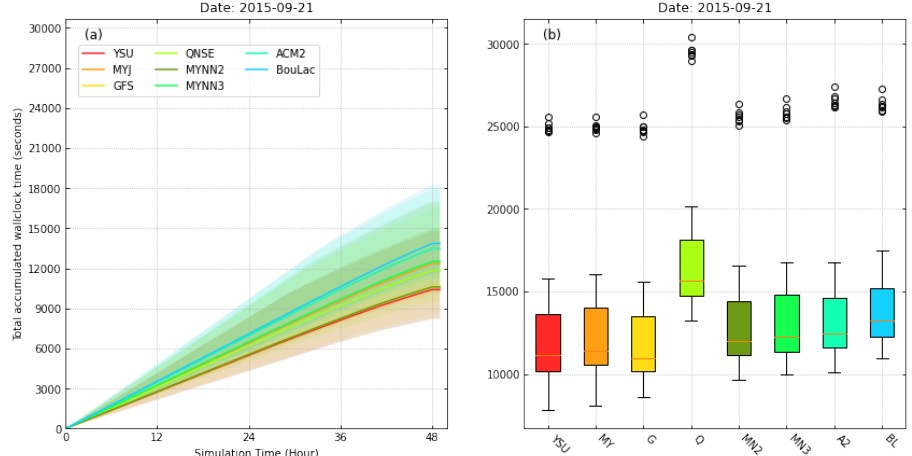

**Figure 5.** The (a) hourly simulation time, (b) total accumulated simulation time, and (c) boxplot by different microphysics configuration within each simulation hour on 2015-09-21. The upper and lower boundaries of the color fill indicates the maximum and minimum simulation time by other physics configurations.

## 4.2 Sensitivity analysis on the clusters of microphysics, PBL, and cumulus parameterization

The 672 ensemble simulations are performed with the target dates and times from section 3.1. The heat maps in Figure 6 and Figure 7 represent two different results based on the cloud cover with high and low variabilities, respectively. Both figures indicates the resulting Kappa by the microphysics cluster along with the cluster of cumulus parameterization and PBL

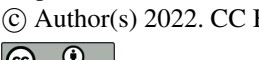

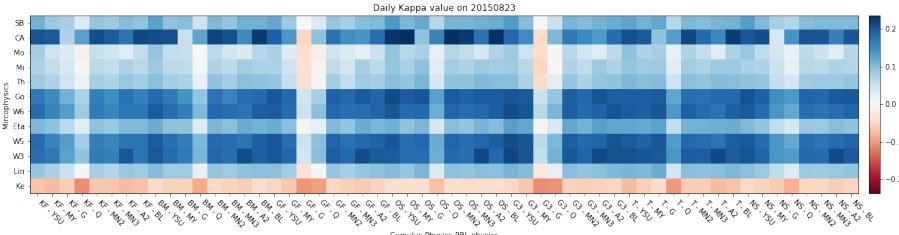

**Figure 6.** The heat map of average kappa for every configuration for the clusters of microphysics (Y axis) with the cluster of cumulus parameterization and PBL (x-axis) on the simulation case "2015-08-23". The data is only use the last 36 hours of simulations for calculating the kappa.

physics. The cloud mask results indicate that the cluster of CAM5.1 outperforms the cloud cover prediction of the other microphysics, but that Goddard and WMS3 also perform well. The kappa indicates that the Kessler microphysics predicted the worst cloud cover overall, irrespective of the PBL physics or cumulus parameterization used. The combination of the three cumulus parameterization Grell-Freitas, Grell-3, and New-SAS, and the PBL physics GFS diminishes the prediction of the
cloud cover. The results from the other four cases can be found in Figure S1-S4.

The simulation results used for comparison with the satellite data exclude the first 12 hours, as the model is runs in weather forecasting mode and hence the first 6-12 hours of simulation hours can be considered to be the spin-up time (Jankov et al., 2007; Kleczek et al., 2014). Only the last 36 hours of the simulation output are therefore used to compare with the satellite data.

The overall results for Kappa from the six test cases (Figure 8) confirm that CAM5.1 performs best for cloud cover, and that both WRF single moments 3 (WSM3) and the Goddard microphysics also perform well. The only exception is simulation case 2015-04-13, where the Goddard microphysics cluster outperfomed the other microphysics. The simulations for case 2015-06-17 and 2015-08-23 are performed well by the Goddard microphysics and the WMS3/5/6 microphysics clusters, where the cloud cover varies more than other cases. The performance of the Goddard microphysics and the WMS3/5/6 microphysics
clusters are equal to the CAM5.1 microphysics cluster when the cloud cover variation is large, as in cases 2015-06-17 and 2015-08-23.

Figure 9 and Figure 10 show the average cloud fraction from hour 12 to 48 from both the satellite data and the simulation result for 2015-08-23 and 2015-09-21, respectively. In this simulation case, we use the different microphysics and the cumulus parameterization of Grell-3D and the PBL physics of MYNN3. In the 2015-08-23 case, the sky is partially clear above the North
Sea and in Eastern Europe, and rather cloudy around the Alps. From the selected microphysics, Kessler, WSM6, Goddard, CAM5.1, and SBU–YLin, different cloud fraction conditions are shown. WSM6, Goddard, and SBU–YL provide a good simulation of the clear sky above the North Sea and Eastern Europe, while the sky above the Alps is as cloudy as the satellite data. The worst case, Kessler, shows a large cloud cover condition over the Eastern Europe, which contradicts the observation. The high variability of cloud cover causes a less cloudy day in case 2015-08-23.





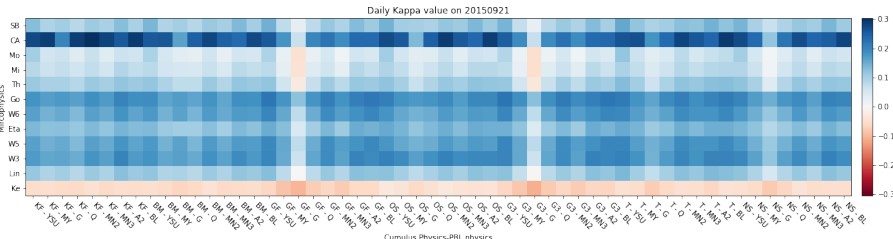

**Figure 7.** The heat map of average kappa for every configuration for the clusters of microphysics (Y axis) with the cluster of cumulus parameterization and PBL (x-axis) on the simulation case "2015-09-21". The data is only use the last 36 hours of simulations for calculating the kappa.

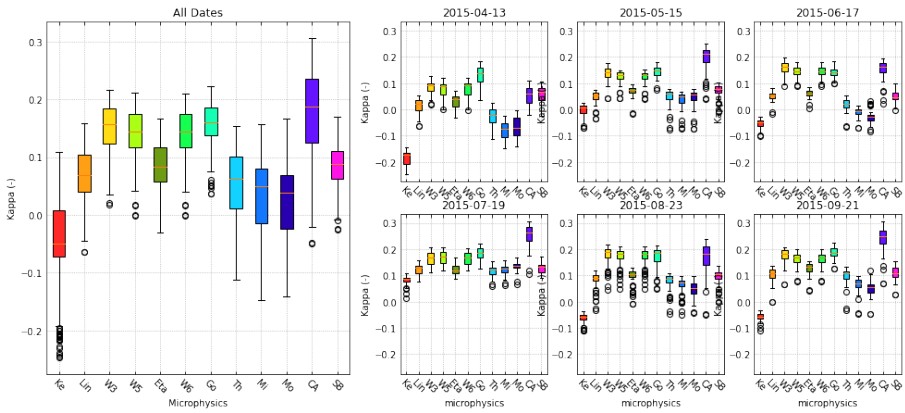

**Figure 8.** The boxplots of kappa in (a) all the simulation dates, (b) case 2015-04-13, (c) case 2015-05-15, (d) case 2015-06-17, (e) case 2015-07-19, (f) case 2015-08-23, and (g) case 2015-09-21.

Case 2015-09-21 is a cloudy day with low variability in cloud cover. A band of clear sky occur above Austria, Slovakia, southern Poland, and Ukrain (Figure 10). The WSM6 and Goddard microphysics simulate less cloud over the clear sky band but produce less cloud cover overall within the model domain. Kessler simulates a cloudy condition over Central Europe.

In Figure 11, the KDE of the cloud cover shows the probabilistic distribution of the average cloud cover for the 36 hours of simulation after hour 12 for case 2015-08-23 and case 2015-09-21. In case 2015-08-23, CAM5.1 microphysics works well for the Kappa but overestimates the cloud cover. The overestimation of cloud cover causes CAM5.1 to perform worse than in cases 2015-05-15, 2015-07-19, and 2015-09-21. Eta, WSM3, WSM5, WSM6, and Goddard microphysics shows similar trends of cloud cover distribution as in the satellite image. In case 2015-09-21, the clear-sky condition (< 5% of cloud cover) is captured wellby CAM5.1, but the cloud cover distributions of the entire CAM5.1 cluster differ from that in the satellite image.

By comparing the result with the microphysics cluster, PBL physics cluster, and cumulus parameterization cluster, we can see the good performance by CAM5.1, WSM3, and Goddard microphysics, while the PBL physics and cumulus paratermeterizations have a secondary impact on the simulation of cloud cover. To obtain a comprehensive result on cloud cover, we





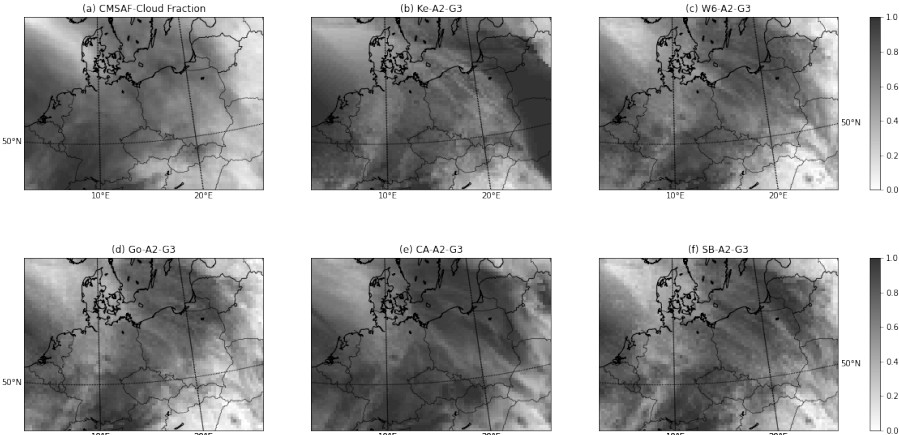

**Figure 9.** Average cloud cover fraction for 36 hours of (a) satellite data and simulation by different microphysics including (b) Kessler, (c) WSM6, (d) Goddard (e) CAM5.1, and (f) SBU–YLin. All simulations are configured with Grell-3D and ACM2 PBL physics, which performs a more skilled prediction than any other combination on 2015-08-23.

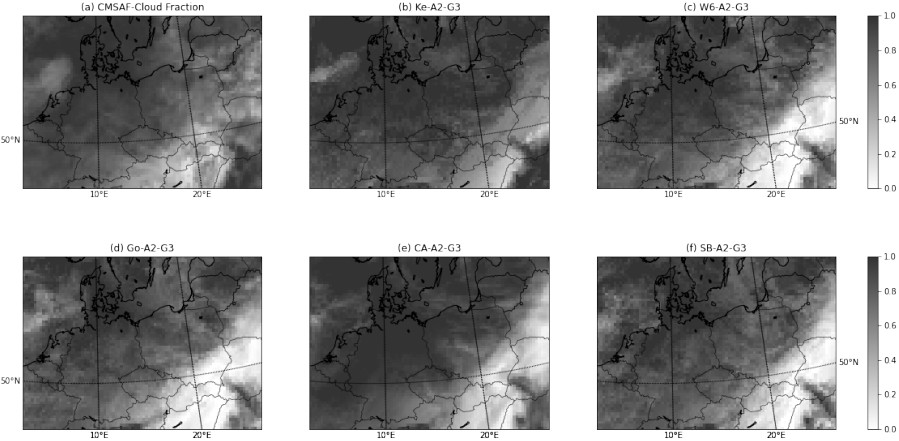

**Figure 10.** Average cloud cover fraction for 36 hours of (a) satellite data and simulation by different microphysics including (b) Kessler, (c) WSM6, (d) Goddard (e) CAM5.1, and (f) SBU–YLin. All the simulations are configured with Grell-3D and ACM2 PBL physics, which performs a more skilled prediction than any other combination on 2015-09-21.

investigate the impact of using different microphysics on the average cloud cover distributions. However, the average cloud covers provide less information and less variability over time. Therefore, the simulated average cloud covers can be performed well and captured by the cloud cover distribution, but the simulation skill can only be captured pixel-by-pixel with the Kappa
score.



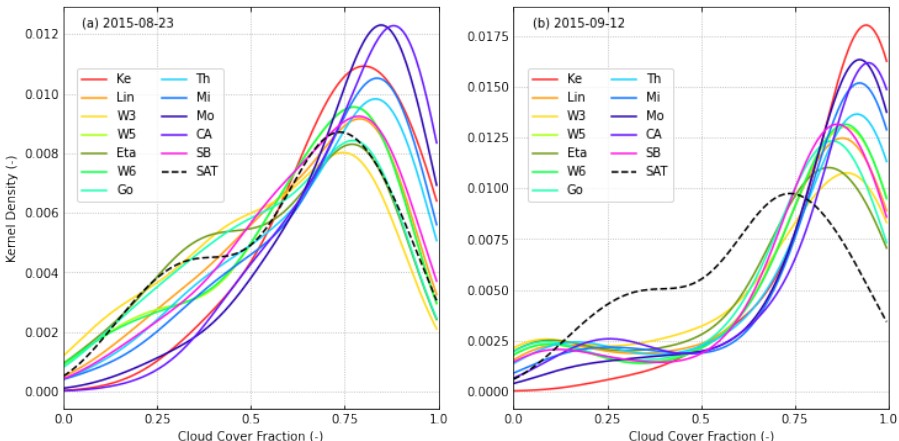

**Figure 11.** The probabilistic function from the kernel density estimation (KDE) of the average cloud fraction for the last 36 hours simulation for (a) case 2015-08-23 and (b) case 2015-09-21. The PBL physics is by ACM2 and the cumulus parameterization is by Grell-3D. The solid color lines depict the KDE from the simulations and the black dash is the KDE from the satellite data

## 4.3 Sensitivity analysis on the clusters of microphysics, PBL, cumulus, radiation schemes, surface layer physics, and land surface parameterization

Additional components to the physics configuration includes different longwave and shortwave radiation physics and land surface layer physics. However, there are more than 1,000,000 combinations from all the physics options. Therefore we narrow

down the choice of the microphysics, PBL physics, and cumulus parameterization from Section 4.2. Accounting for the support of the simulation of the graupel mixing ratio for ESIAS-chem, we predominantly use the microphysics of WSM5, WSM6, and Goddard. CAM5.1 performs the best across 5 of the 6 test cases but it is not included because of the high cost of simulation resources with the same CPU settings. MYNN2, MYNN3, and ACM are selected because of their good performance with the selected microphysics. From the heatmap (Figure 6 ) both Grell 3D and Tiedtke work well with Goddard and WSM6. We also

choose Kain-Fritsch, which is widely used (e.g. Warrach-Sagi et al. (2013) and Knist et al. (2017)), for comparison. The PBL physics by MYNN2, MYNN3, and ACM2 perform well across all the simulations and is chosen for this simulation case.

     Figure 12 shows the heat map of the simulation case 2015-09-21 and the physics configuration of Goddard and ACM2. The Goddard radiation schemes perform skillful predictions of cloud cover. This heat map also indicates good combinations of microphysics, cumulus parameterization, and radiation schemes as well as combinations of PBL physics, surface physics and land surface models by row and column, respectively. By row, the Goddard works overall with the Tiedtke and Grell-3D

cumulus parameterization over all. By the column, the heat map shows that ACM2 PBL physics can improve the simulation with all the microphysics but with less improvement for the radiation schemes RRTM and Dudhia. Under the same condition, MYNN3 with Grell-3D and RRTM and Dudhia perform better with different microphysics.From the 513 combinations of physics configurations, the range of the Kappa score is between 0.15 and 0.24. The microphysics and the PBL physics are



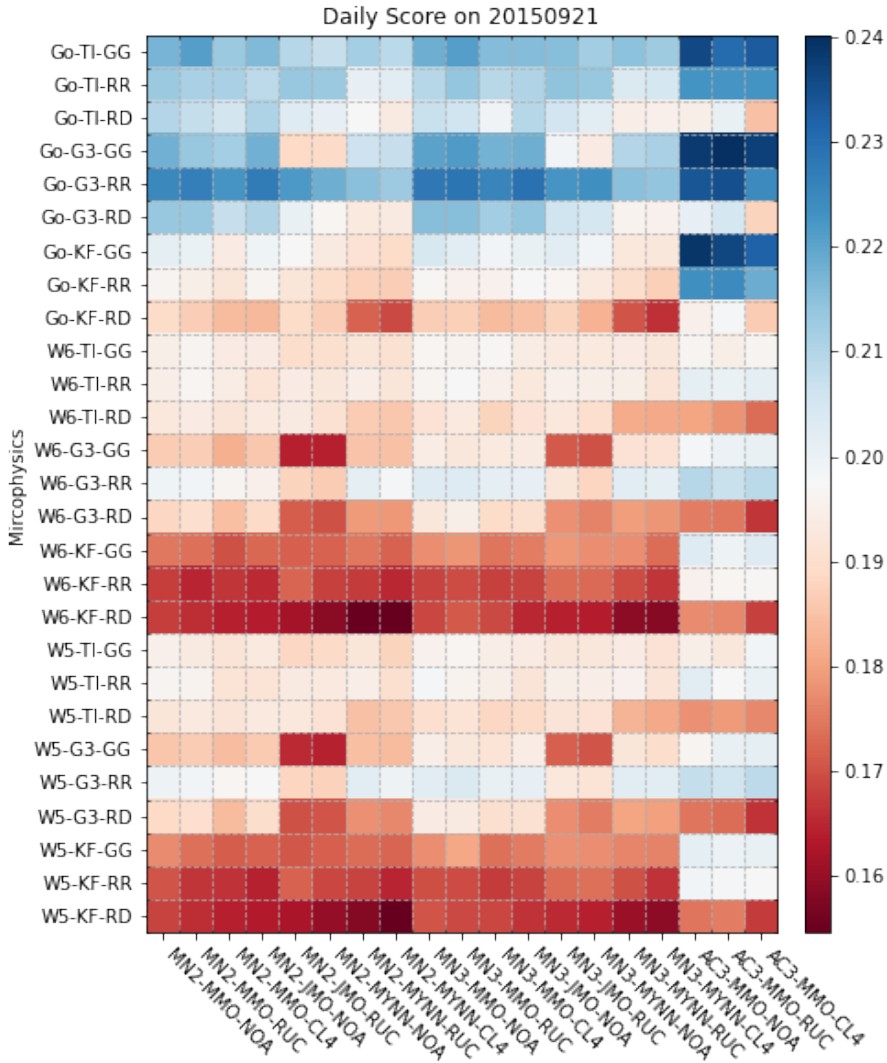

**Figure 12.** The heat map of the average kappa for the configuration for the clusters of microphysics, cumulus parameterization, and the radiation physics (Y axis) with the cluster of PBL-physics, surface physics, and the land surface model (x-axis) on the simulation case 2015-09-21. The data is only use the last 36 hours of simulations for calculating the kappa.

chosen from the results of section 4.2, which is why the improvement is not significant compared to the improvement from changing either the longwave and shortwave radiation scheme, or the surface layer physics, or the land surface models.

In case 2015-06-17, the cloud cover distribution has a very high variability, and therefore the simulation skills increase their variabilities in Kappa with different combinations. When all the cluster of microphysics with Kain-Fritsch perform less than 0.1, the simulations are better with the combination of MYNN2 and MYNN and RUC; the combination of MYNN3, MYNN,

and Noah; and the combination of ACM2, MMO, and CLM4. The pattern of outperforming Kappa is also shown in the result



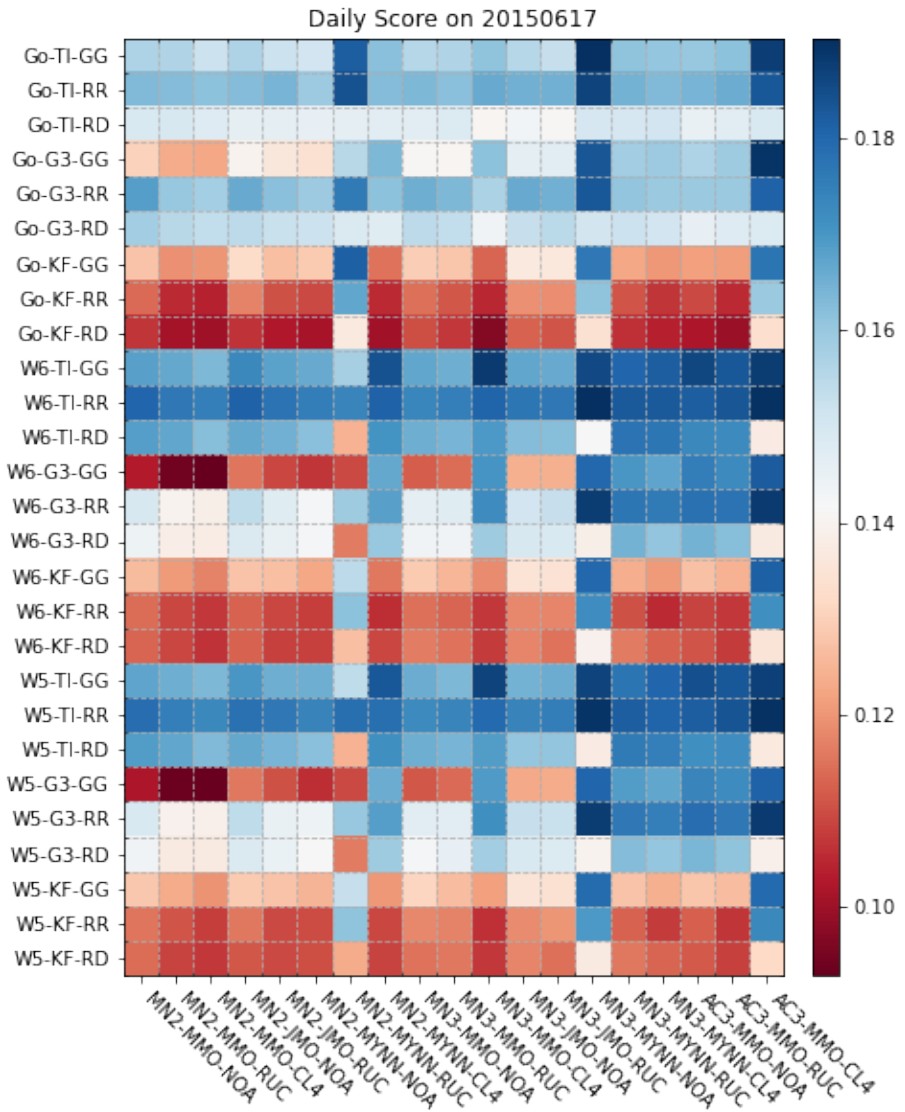

**Figure 13.** The heat map of the average kappa for the configuration for the clusters of microphysics, cumulus parameterization, and the radiation physics (Y axis) with the cluster of PBL-physics, surface physics, and the land surface model (x-axis) on the simulation case 2015-06-17. The data is only use the last 36 hours of simulations for calculating the kappa.

for 2015-07-19 (Figure S7), which includes the worst performance of Kappa among the six cases. The results from the other four cases can be found in Figure S5-S8.

The boxplot (Figure 14 ) shows the overview of the six cases along with the cluster of microphysics and the cumulus parameterizations. The cluster of microphysics and cumulus parameterizations show less variability and are very similar for each case, which indicates that high variability occurs in other clusters of physics. The maximum of the boxplot





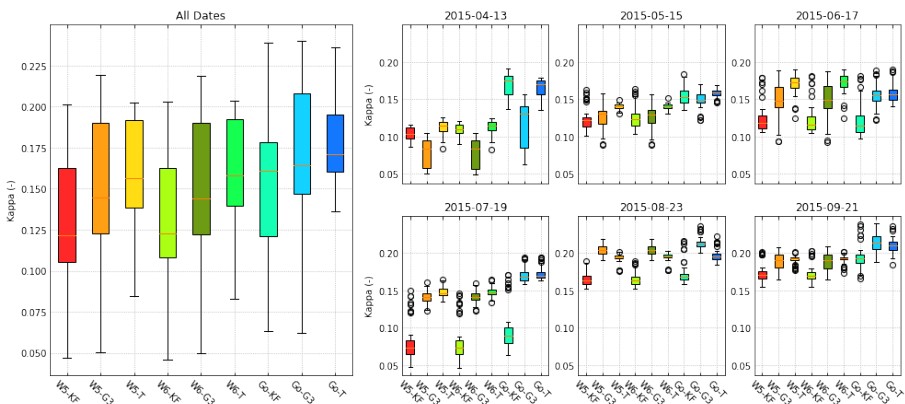

**Figure 14.** The boxplots of kappa from microphysics and cumulus parameterization clusters in (a) all the simulation dates, (b) case 2015-04-13, (c) case 2015-05-15, (d) case 2015-06-17, (e) case 2015-07-19, (f) case 2015-08-23, and (g) case 2015-09-21.

(3rd quantile $+ 1.5 \times$ interquartile range) shows that the combination with the Grell-3D cumulus parameterization can achieve maximum average Kappa scores. Using Goddard microphysics and Tiedke cumulus parameterization can be less variable than the ones using other physics or parameterization, and its median Kappa indicates that this combination can outperform other microphysics and cumulus parameterization clusters. The boxplots from all the cases show that the cumulus parameterization, which is more advanced than the Kain-Fritsch can improve the Kappa. The resulting Kappa is not significantly different between using Grell-3D and Tiedtke.

### 4.4 Sensitivity analysis on the clusters of microphysics and PBL for its impact on stochastic simulation

Stochastic weather forecasting depends on a large ensemble size of simulation. To study the impact of physics configuration on the stochastic simulation, we generate $31 + 1$ ensemble members to simulate 48 hours of stochastic weather forecasting. The total cloud fractions after 12 simulation hours are used to analyse the effect of different physics configurations on the simulation result and its impact on the probability. The simulation is conducted for the six cases with the same domain setting and input data and the same cloud data are also used.

Figure 15 shows the probabilistic cloud cover fraction within the $25^{th}$ to $75^{th}$ percentiles and $5^{th}$ to $95^{th}$ percentiles of the simulations. The developement of the mean cloud cover fraction are compared to the mean cloud cover fraction of the satellite data. The *rmse* and standard deviation are used to the show the comprehensive result from the temporal development of cloud cover within the last 36 hours of simulation.

The Kessler microphysics show an overestimation of cloud cover fraction and the largest *rmse* of all microphysics. Moreover, the two peaks of cloud cover fraction are not captured at the $12^{th}$ hour and the $36^{th}$ hour of simulation. In addition to Kessler microphysics, the microphysics cluster can capture these two peaks with the ACM2 physics. The smallest *rmse* is performed by SBU–Ylin microphysics and ACM2 PBL physics and the second smallest by the WSM6 microphysics and ACM2 PBL physics. The biggest standard deviation is produced by SBU–Ylin and MYNN2 PBL physics, and this physics configuration therefore



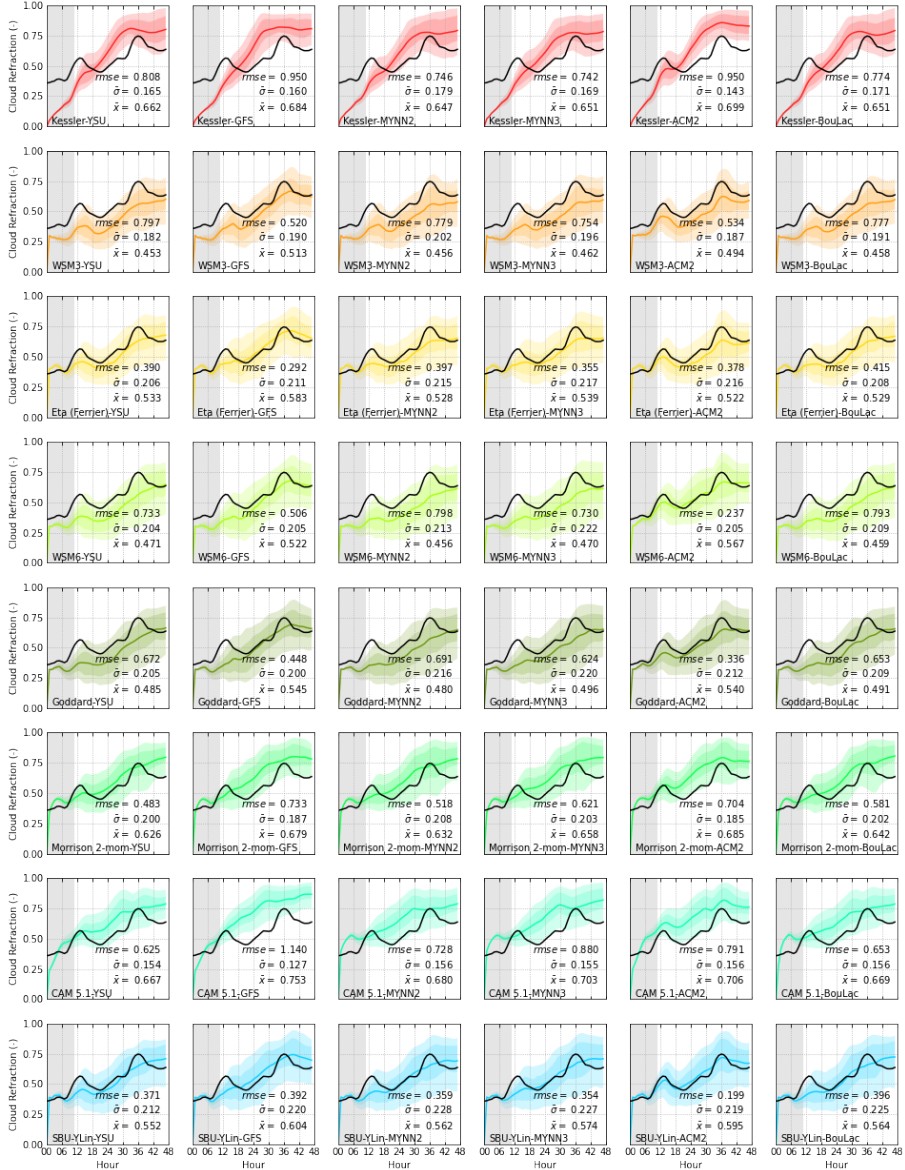

**Figure 15.** The mean cloud cover fraction (-) the observation from the satellite (black line) and by simulation (color line) from the combination of microphysics (by different color) and PBL physics (in different row) in case 2015-08-23. The color block represents the range of percentiles, the darker block is limited between 25% and 75%, and the lighter block is lmited between 5% and 95%. The grey block indicates the spin-up time for ESIAS-met, which is not included in the root mean square error *rmse*, standard deviation ($\bar{\sigma}$), and the mean simulated cloud cover fraction ($\bar{x}$).

produces the largest spread of possible results. In the 2015-09-21 case (Figure 16, the WSM6 microphysics shows the greatest spread with the largest standard deviation. In both cases, the CAM5.1 microphysics combined with the GFS PBL physics

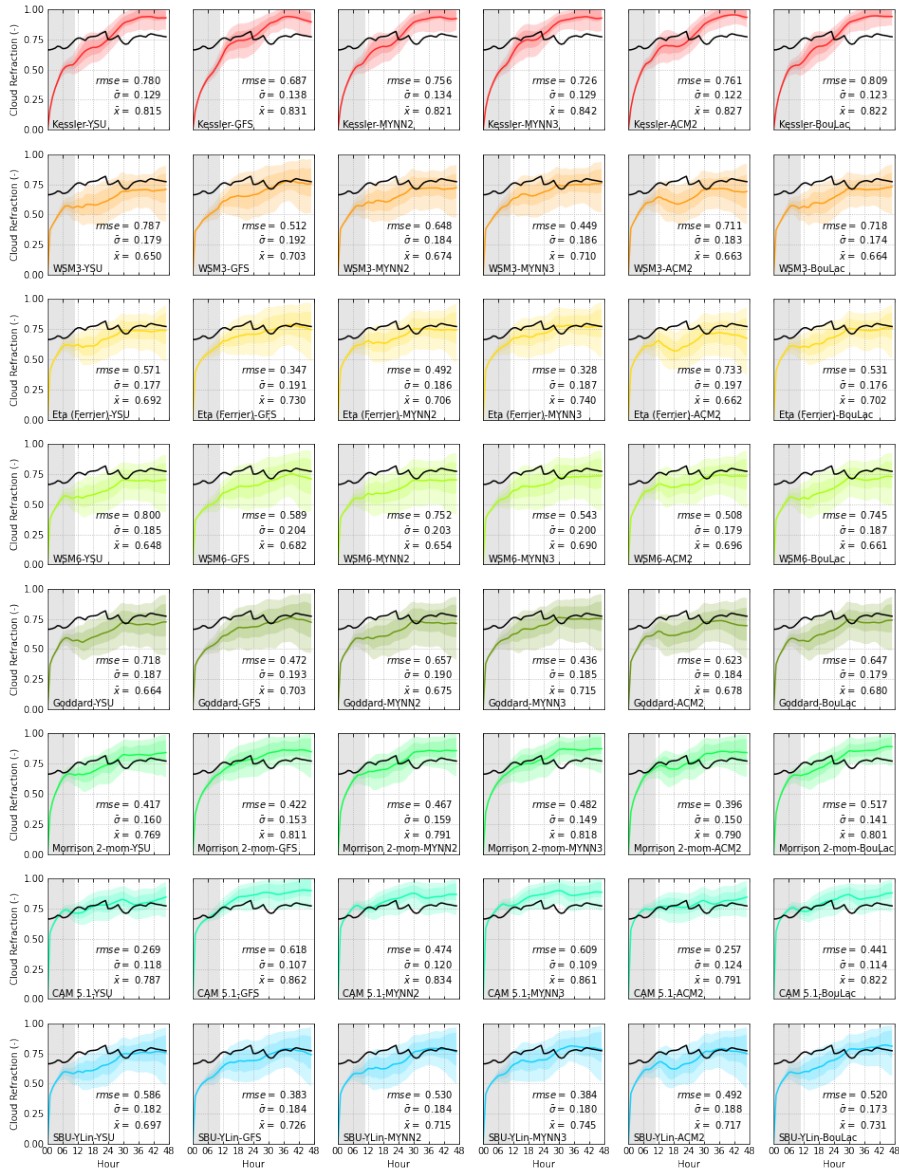

**Figure 16.** The mean cloud cover fraction (-) the observation from the satellite (black line) and by simulation (color line) from the combination of microphysics (by different color) and PBL physics (in different row) in case 2015-09-21. The color block represents the range of percentiles, the darker block is limited between 25% and 75%, and the lighter block is limited between 5% and 95%. The grey block indicates the spin-up time for ESIAS-met which is not included in the root mean square error *rmse*, standard deviation ($\bar{\sigma}$), and the mean simulated cloud cover fraction ($\bar{x}$).

produces the smallest probability distribution of cloud cover fraction. The color blocks show the simulation skill in capturing

the cloud cover fraction within a certain percentile. The MYNN2, MYNN3, and ACM2 PBL physics not only produce a larger





probabilistic distribution than the other PBL physics, but also perform better results with the WSM6, Goddard, and SBU–YLin microphysics. The results from the other four cases can be found in Figure S9-S12.

## 5    Discussion and conclusion

### 5.1    Impact of physics configuration on the simulated cloud cover

The Kappa heatmap from the cloud cover masks shows that microphysics is the key aspect affecting the simulation of cloud cover in the European domain. The Kappa values show a consistently good result by using the WSM3, WSM5, WSM6, Goddard, and CAM5.1 microphysics based on the six simulation cases and 672 combination of physics configuration (Set 1) and 513 combination of physics configuration (Set 2). The good performance in predicting cloud cover fraction using WSM6 had been previously explained by Pieri et al. (2015) and Jankov et al. (2011).

The employment of ACM2, MYNN2, and MYNN3 PBL physics can lead to good results in the cloud cover mask. For the six cases the Morrison and Kessler microphysics schemes should be avoided, and the QNSE and GFS PBL physics should also be avoided. The research results of Borge et al. (2008) point to the same choice of WSM6 microphysics, but our results instead suggest using ACM2, MYNN2 and MYNN3 PBL physics, while Borge et al. (2008), Gbode et al. (2019) and Stergiou et al. (2017) report the YSU or MYJ PBL physics to be the better physics configuration. However, Borge et al. (2008) focus on the

prediction of wind properties, temperature, and humidity over Spain for 168 hours of simulation; Gbode et al. (2019) focus on the prediction of precipitation over West Africa during the monsoon period, and Stergiou et al. (2017) focus on the prediction of temperature and precipitation over Europe during two different months.

The boxplots by different physics combinations illustrate that the employed land surface model or radiation schemes are not as consequential as the change of configuration for microphysics and PBL physics. However, the results from the Set 2

sensitivity analysis showed significant differences for different cumulus parameterizations, although not as large as from the change of the microphysics and PBL physics, and the Kappa values are generally less for the Kain-Fritsch than from Grell-3D and Tiedtke.

### 5.2    Impact of physics configuration on the result of stochastic simulation

ESIAS-met is the ensemble version of WRF, and therefore it is very important to understand the impact of different physics

configuration on the stochastic results. We identify the most sensitive physics clusters and use the physics configuration (Set 3) to generate 1,536 ensemble members. However, we only simulate $31 + 1$ ensemble members due to the limitation of computation resources (i.e. a single case uses 18,432 cores for a total of 1,536 ensemble members).

The results of the stochastic simulations show that the combination of physics highly affects the probability distribution. Kessler overestimates the cloud cover fraction. Similar to the results from the physics configuration sensitivity analysis, the

cloud cover prediction is simulated well by the microphysics CAM5.1, Goddard, and WSM6. Moreover, the SKEBS stochastic scheme produces broader probabilistic distributions by employing WSM6 and Goddard and therefore the average cloud cover





fraction can be captured by these microphysics. The CAM5.1 microphysics produces the most accurate results when compared pixel by pixel, but the probabilistic distribution is smallest compared to the other microphysics.

The PBL physics not only change the development of cloud cover fraction but also affect the probabilistic distribution of the
cloud cover fraction. The GFS and MYNN2 scheme produce less dynamic cloud cover and thus produce higher *rmse* values. The ACM2 produces a more dynamic development of cloud cover, but its probabilistic distribution is slightly less than that of MYNN2 and MYNN3.

Stochastic analysis shows a contradiction between the deterministic simulation and the probabilistic simulation. The most suitable configuration for an unbiased deterministic forecast may differ from that of the best multi-member ensemble forecast
that best captures the uncertainty and diversity of the possible outcomes. For the most accurate simulation, the CAM5.1 microphysics and the ACM2 PBL physics performed best, while SBU–YLin with MYNN2 shows better results in terms of mean cloud cover and the observed satellite data. Of all the six cases, the CAM5.1 microphysics produces the least probabilistic distribution, while Goddard and WSM6 microphysics can generally produce broader probabilistic distributions.

Jankov et al. (2019) and Li et al. (2019) both report an insufficiency of ensemble spread with stochastic schemes (e.g.
SKEBS or SPPT) and suggest using a multiphysics simulation to obtain a greater spread. In our simulations, the multi-physics can enhence the spread of ensemble simulation, though the ensemble spread by multi-physics simulation is based on the uncertainty in the model physics. A probabilistic simulation is a solution that predicts weather forecasting from the distribution of large ensemble member sets, but we should also consider the accuracy of the model physics. The multi-physics simulation serves as an ensemble simulation based on random estimation. For instance, different model physics simulate different cloud
patterns, and the sum of two cloud patters then eliminate cloud boundaries and are not in agreement on the resulting front. Moreover, Jankov et al. (2019) performed a stochastic simulation with only four ensemble members; the insufficient number of ensemble members may be the root cause of the small ensemble spread reported in their results.

## 5.3 Choice of physics configurations

The simulation results do not indicate a single best option for the physics configuration. Many studies focusing on very different
topics, including different resolutions (Warrach-Sagi et al., 2013; Pieri et al., 2015; Knist et al., 2017, 2018), inputs (Pieri et al., 2015), microphysics (Jankov et al., 2011; Rögnvaldsson et al., 2011), PBL physics (García-Díez et al., 2013), cumulus parameterizations (Gbode et al., 2019) , land surface models (Jin et al., 2010) , and the combination of different physics (Borge et al., 2008; Santos-Alamillos et al., 2013; Awan et al., 2011; Jankov et al., 2007; Pieri et al., 2015; Stergiou et al., 2017; Otkin and Greenwald, 2008; Li et al., 2019; Varga, 2020). However, these studies focus on different target variables
and meteorological states with different weather forcing input, observation data, study areas, and time scales, which therefore produce very different results for the choice of physics or parameterization. Our simulation results do not give a clear indication on the meteorological aspect across temporal and spatial scales. Therefore we offer a recommendation on the choice of physics configurations for studying the European domain and for weather forecasting purposes. However, further investigations must be carried out for more comprehensive insights into on the spatial scales, more combinations of physics configurations, and
different input data.



Nevertheless, the simulation applying 20 km horizontal resolution with the day-ahead weather forecasting, and the employment of WSM6, Goddard, and CAM5.1 microphysics improve deterministic weather forecasting on cloud cover, while the employment of WSM6, Goddard, and SBU–YLin microphysics benefit the stochastic weather forecasting on cloud cover with a greater probabilistic distribution. The PBL physics are better simulated by ACM2 and MYNN3. However, cumulus parame-
terizations, surface layer physics, and the land surface models do not increase the accuracy significantly. A more comprehensive study should include some promising combinations of physics configurations, and include the short- or long-term effects of applying different physics at different spatial scales, such as continental or global scales, and include a 1 km resolution to study the dynamics and local conditions at convection-resolving scale.

The performance of probabilistic simulations relies on their probabilistic distribution in addition to their accuracy. The
SBU–YLin, Goddard, and WSM6 schemes generate broader probabilistic distributions, while Kessler and CAM5.1 generate the narrowest. The PBL scheme also has a significant effect on the probabilistic distributions; MYNN2 and MYNN3 generate wider distributions while GFS generates smaller ones. The best combination for probabilistic simulations ideally has accurate ensemble means and realistically broad distributions. The Goddard, WSM6, or SBU–YLin microphysics with MYNN3 are potential choices. When the mean of the simulations is close to the mean of the observation, the reality can be better captured.
However, as mentioned by Sillmann et al. (2017), the technique of scoring ensemble simulations remains a challenge in better estimating the results from the view of probabilistic analysis.

### 5.4   Future work

We performed simulations without a nested domain for a higher resolution simulation, which might be useful for investigating the effect of the resolution on multi-physics for convection-resolving simulations. Exascale high-performance computing might
enable make such studies for scientific research and provide an opportunity for investigating the scalability of ultra-large ensemble simulation systems (Neumann et al., 2019; Bauer et al., 2021). To this end, the ESIAS system presented here has been designed to perform data assimilation with the advantage of the elastic ensemble simulation frameworks. Further development will focus on data assimilation, such as the use of the particle filter with the particle removal function (van Leeuwen and Jan, 2009).

## 6   Conclusions

This study introduces an ensemble simulation system for conducting ultra-large ensemble simulations in Europe and with multiphysics simulations and probabilistic simulations. We use the meteorological part of the system to perform simulations and generate large ensembles to perform sensitivity analyses on the effect of various physics configurations on the cloud cover fraction. The simulation is conducted within Europe without any nested domain for a day-ahead forecasting (48 hours) for six
days. The sensitivity analysis is based on 672 and 513 combinations of physics configurations, investigating combinations of three and six different physics elements, respectively. Additionally, we perform a sensitivity analysis to determine the effect of



...

 




physics configurations on cloud fraction and use the calculation of Kappa to identify the score of the match rate by cloud cover masks.

The sensitivity analysis of the combination of three physics configurations – microphysics and the planetary boundary layer (PBL) physics and the cumulus parameterization – shows that the microphysics has the greatest influence on the cloud cover. The Goddard, WSM3, and CAM5.1 microphysics consistently perform better than the other microphysics, but the amount of computation time required for CAM5.1 is relative high. The Goddard and WSM3 scheme increase performance when the cloud cover is more dynamics. The PBL physics also has a significant effect on the results and shows better agreement with YS and ACM2, but less agreement with GFS and QNSE PBL physics.

The sensitivity analysis on the combination of six physics configurations – including microphysics and the PBL physics, the cumulus parameterization, longwave and shortwave radiation schemes, surface layer physics, and land surface models – shows that the microphysics affects the cloud cover the most, and that the ACM2 PBL physics significantly increases the accuracy of predicting cloud cover. However, the physics configurations of surface layer physics and land surface models are not as significant as other physics.

The sensitivity analysis on the combination of two physics configurations and its effect on stochastic simulation shows a significant effect on the probabilistic distributions on the cloud cover fractions. WSM6 and SBU–YLin microphysics with MYNN2 and MYNN3 capture the cloud fraction better within the greater range of their probabilistic distributions than have the other models, although the WSM6 and SBU-YLin with ACM2 better captures the dynamics of cloud fractions when the cloud clover has more variability during the simulation time.

The simulation results indicate a pathway for improving model physics and demonstrate the potential of ultra-large ensemble simulations and high-performance computers approaching exascale. The multi-physics simulation however produces a larger ensemble spread compared to the stochastic schemes, although the result from the sum of the multi-physics may not be realistic. The employment of ultra-large ensemble simulations with suitable physics configurations can improve both accuracy and the probabilistic distributions from simulation results.

*Code availability.*

The codes of ESIAS-met and also the pre- and post-processed codes are available for public domain via https://zenodo.org/record/6637315#.YqbhgBxBzeK (DOI:10.5281/zenodo.6637315). The modelling and analysis tools can be found in the code repository: https://github.com/hydrogencl/WRF_TOOLS and https://github.com/hydrogencl/SciTool_Py.

**Appendix A**





**Table A1.** Abbreviation of the physics employed in this study for microphysics, cumulus parameterization, and planetary boundary layer (PBL) physics.

| Microphysics | | Cumulus Parameterization | | PBL physics | |
|---|---|---|---|---|---|
| Full name | Abbr. | Full name | Abbr. | Full name | Abbr. |
| Kessler | Ke | Kain-Fritsch | KF | YSU | YSU |
| Lin (Purdue) | Lin | Betts-Miller-Janjic | BM | MYJ | MY |
| WSM3 | W3 | Grell-Freitas | GF | GFS | G |
| WSM5 | W5 | Simplied Arakawa-Schubert | OS | QNSE | Q |
| Eta (Ferrier) | Eta | Grell-3 | G3 | MYNN2 | MN2 |
| WSM6 | W6 | Tiedtke | T | MYNN3 | MN3 |
| Goddard | Go | New SAS | NS | ACM2 | A2 |
| Thompson | Th | | | BouLac | BL |
| Milbrandt 2-mom | Mi | | | | |
| Morrison 2-mom | Mo | | | | |
| CAM 5.1 | Ca | | | | |
| SBU–YLin | SB | | | | |

**Table A2.** Abbreviation of the physics employed in this study for the combination of shortwave and longwave radiation scheme, surface layer physics, and land surface model.

| Radiation scheme (shortwave & longwave) | | Surface layer physics | | Land surface model | |
|---|---|---|---|---|---|
| Full name | Abbr. | Full name | Abbr. | Full name | Abbr. |
| RRTM & Dudhia | RD | Revised MM5 Monin-Obukhov | MMO | unified Noah | YSU |
| RRTMG & RRTMG | RR | Monin-Obukhov (Janjic Eta) | EMO | RUC | MY |
| New Goddard & New Goddard | GG | MYNN | MYNN | CLM4 | G |



*Author contributions.*

YSL and GG designed the experiments and wrote the manuscript, and YSL performed the simulation. HE is the founder of the idea of ESIAS-met and helps to prepare of the manuscript.

*Competing interests.*

The authors declare no conflicts of interest with respect to the results of this manuscript.

*Acknowledgements.* This work was fully supported by the Council of the European Union (EU) under the Horizon 2020 Project "Energy Oriented Center of Excellence: toward exascale for energy" - EoCoE II, Project ID 824158. The authors also gratefully acknowledge the Earth System Modelling Project (ESM) for funding this work by providing computing time on the ESM partition of the supercomputer JUWELS at the Jülich Supercomputing Centre (JSC).





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
