# Peer review of "Optimization of weather forecasting for cloud cover over the European domain using the meteorological component of the Ensemble for Stochastic Integration of Atmospheric Simulations version 1.0"

_Geoscientific Model Development, 2022_

## Author Comment (AC1)

**Authors' response to reviewer 1: Optimization of weather forecasting for cloud cover over the European domain using the meteorological component of the Ensemble for Stochastic Integration of Atmospheric Simulations version 1.0**

Yen-Sen Lu[1,4], Garrett Good[2], and Hendrik Elbern[3]

[1]Institute of Energy and Climate Research – Troposphere (IEK-8), Forschungszentrum Jülich GmbH, 52425 Jülich, Germany
[2]Fraunhofer Institute for Energy Economics and Energy System Technology IEE, Königstor 59, 34119 Kassel, Germany
[3]Rhenish Institute for Environmental Research at the University of Cologne, Cologne, Germany
[4]Jülich Supercomputing Centre, Forschungszentrum Jülich, Jülich, Germany

**Correspondence:** Yen-Sen Lu (ye.lu@fz-juelich.de)

First of all, we authors thank the insight and detailed review by the reviewer 1. Those valuable suggestion will be taken into account for overhauling the manuscript. Please see the replies to the following sections.

**1 Reply to the *Major Comments**

*The overall manuscript lacks clarity, making it very difficult for the reader to understand the experiment design, analysis, and results. At times, the text is also missing critical justification for decisions made by the authors.*

The language has been overhauled by a native speaker for readbility and we have addressed the comments below in the revision. We think the new text better communicates the methodology, in pacticular, the iterative approach to the sensitivity analysis.

*It is not very clear why the authors chose to separate the experiments into the three sets described. To help clarify, it would be useful if each set of experiments listed all physics parameterizations used, with the "cluster" name referenced, so they can be compared more easily. Also, the ensemble members that use stochastic physics need to be clearly identified. The authors mention using SPPT as well as SKEBS, but only SKEBS appears to be described in 2.2.*

We have clarified in the paper the iterative approach of starting with a limited study of a very large number of physics due to computational resources and narrowing this down to fewer physics studied with more detailed simulations.

20    We've updated the language and also the table of the listed physics configuration to include the clusters, and thus the reader should be able to understand the connection between the clusters and the optional physics.

The ensemble members that use the stochastic physics in this study only employ the SKEBS since it performed better in a previous study. We update this information with Berndt (2018) reporting that ESIAS-met employing SKEBS can produce

25    more instability effectviely than employing the SPPT scheme. The SPPT scheme exists in the ESIAS-met but we did not used it to conduct the stochastic members in the simulation. We've clarified this in the manuscript.

*It is not clear if the authors considered the appropriateness of each physics parameterization for the resolution used in the simulations. Certain physics parameterizations are targeted at specific resolutions, unless they are truly scale-aware. Some*

30    *parameterizations used in WRF are specifically targeted toward convection-allowing scales ( 3 km resolution). Therefore, the authors need to thoroughly explain which schemes may not be appropriate for the 20-km resolution they are running, if any, and exclude those from their runs.*

We are grateful for raising this important point. From the user's guide (WRF, 2015), Eta microphysics and Morrison are listed

35    as only for the finer resolution ($< 5km$) and cloud-resolving simulations, respectively. We've thus removed these two microphysics from the further results and commented on their exclusion in the model setup section. All the plots in the manuscript and the supplementary material are updated accordingly.

*Additional details about the ESIAS and general experiment design would be helpful. For example, why does ESIAS use 1000*

40    *members? Does it always run with that many? If it is specifically being run with 1000 members only in this case to sample as many WRF physics parametrizations as possible, please mention that. Which of the 1000 members employ stochastic physics? Is the ESIAS always run at 20 km resolution using the same 180 x 180 grid point domain? Can it be configured to run differently? Is it run operationally?*

45    We have reworked the description in section 2.1. Acknowledging the fact that today's operational ensemble systems at weather centres operate with ensembles sizes < O(100) members, ESIAS is designed to offer the option of ensemble sizes at least one order of magnitude larger, that is O(1000). ESIAS-met does not necessary to run on 1,000 members, but is a flexible system capable of such runs. For instance, in our simulations it was initially run for 672 members in 48-hr simulaitons in Set 1, but also 4 members for a year of simulation in Set 4. The `WRF_TOOLS` can be employed to produce the needed input namelist

50    without working one by one. For the set 1 and set 2 there is no member employing stochastic physics. ESIAS-met can be configured like the normal WRF to employ different nested-domain simulation or other requested time or spatial resolution. ESIAS is further designed to allow for alternative ensemble member definitions in the future, mainly envisioning further flexibility for various SPPT approaches. For our project, we have performed the simulation over the Europe and North Africa to study the sensitivity analysis on the Saharan dust events (unpublished). Our model can be also run operationally to produce

55    weather forecasting if needed.

*Section 3.1 is unclear. GEFS data are apparently used, but is it for both ICs and LBCs? What is the frequency at which LBCs are applied? What is the forecast length (48 hours)? What is the frequency of the output (every three hours)?*

60    We hope this is clearer in the reworked text with added details. Yes, the ICs and LBCs of GEFS data are used. The frequency of the LBCs are every 3 hours. The forecast length is 48 hours including the spin-up time. The further used ECMWF ERA5 data for Set 4 (long-term simulation) is also described in the Section 3.1

*The work described in this manuscript may represent one of the largest physics sensitivity studies using the WRF to date*
65    *(certainly this is true for the evaluation of cloud cover?). It may be worthwhile for the authors to highlight this fact in the abstract/conclusions. Grammar should be double-checked throughout the manuscript. Some explanations by the authors are very difficult to understand. There are also numerous typos found throughout the manuscript.*

Thank you for highlighting this, we will add it to the abstract. We also work on checking the grammar and typos.
70

*The authors state in the conclusions that they "... offer a recommendation on the choice of physics configurations for studying the European domain and for weather forecasting purposes." The manuscript only focused on the evaluation of cloud cover, which is just one of tens to hundreds of variables that are important for NWP. If the authors wish to provide physics recommendation for general NWP over Europe, many more variables need to be evaluated.*
75

We completely agree that this study can only offer recommendations for the cloud cover and have clarified the focus on energy meteorology. Evaluating further variables and aspects of the NWP performance would be far more complex and was not within the objectives of our funding. We nevertheless hope that the results and technical work presented here can lead to future studies of more meteorological variables and expect more collaboration with experts in the field of observation and
80    analysis. We have modified the wording to limit the scope of the results.

**2    Reply to the *Minor Comments**

*Lines 41-43: Are the authors saying that most physics combinations will exhibit a bias when compared to surface-based observations? If so, sure, and that's inevitable as model physics will never be completely bias-free. Also, why only surface-based*
85    *observations? Upper-air observations can be used equally to verify model simulations, with model physics having the potential to impact upper-air variables just as much. Model simulations also include more than just physics, so it's not possible to say that all bias is due to just the model physics. In addition, having some kind of bias doesn't necessarily make a physics*

*parameterization or suite "unsuitable for deterministic forecasts". All operational models have some kind of physics bias, and work is always ongoing to minimize the error.*

We totally agree with this statement and now touch on this in the introduction as: *There is thus potential for optimization, as most physics combinations can be expected to be biased as compared to observations.*

*Line 49 - What is "the scientific challenge of proper scoring rules"? Please clarify.*

We have reworded this with other comments.

*Lines 50-51 – Is the "technical challenge" creating "large supercomputing facilities", finding the resources to run on large supercomputing facilities, the ability of an ensemble to forecast extreme and damaging events, or all of the above? Please clarify.*

We clarify as: *Presently, a large supercomputer can produce sufficiently ultra-large ensembles of $O(1,000)$ members (at moderate resolution) if challenges in e.g. I/O performance and MPI communication are addressed. The ESIAS framework (Berndt, 2018; Franke et al., 2022) has been developed to accomplish ultra-large ensemble forecasts of up to $(O)1000$ members, demonstrated in this study with both multi-physics and stochastic schemes for the probabilistic simulation of the cloud condition.*

*Line 52 and elsewhere – A simulation isn't probabilistic by itself, but probabilistic forecasts for a given event can be created from an ensemble forecast. I would replace "probabilistic simulations" with "ensemble-based probabilistic forecasts".*

We update the text with: *ensemble-based probabilistic forecasts*

*Line 54-55 – It isn't clear whether stochastic physics is used in all 1000 ensemble members described in the ESIAS, or whether there is a subset of additional members that employ stochastic physics. Please clarify.*

We have updated the text as mentioned above.

*Line 56 – "cope with" or "meet"?*

The sentence is rephrased and thus this error is gone.

*Line 59 – "ESIAS-met" has not been defined yet. It appears to be defined in line 63, but it's not clear what the difference is between ESIAS and ESIAS-met until the next section.*

125    We have modified the manuscript to introduce ESIAS-met in the next session and thus keep using *the meteorological component of ESIAS* in the introduction.

*Lines 59-62 – Please double check grammar. Also, are multi-physics simulations combined with stochastic simulations?*

130    The clarified these details in section 2.2

*Line 78 – "to better fit the system" – I'm not sure what the authors are trying to say here. Any specific reason why only SPPT and SKEBS are used? Did the authors also look at using SHUM or SPP?*

135    We have rephrased the sentence as: *The namelist generation by the WRF TOOLS of ESIAS-met is the same as for WRF V3.7.1 and the input and output filenames are flexible.*

We only use SPPT and SKEBS because of it's original inclusion in the ESIAS system. The SPP can be used if ESIAS-met is updated to use the WRF4. The SHUM (stochastic boundary-layer humidity) or other stochastic schemes are not considered 140    since the sensitivity analysis of stochastic schemes themselves is not our main focus. However, ESIAS-met is designed to easily adopt both methods, SHUM taken as a variant of SPP(T).

*Line 91 – Can the authors briefly describe the "different approach" used in the other study?*

145    We have extended description with: *which is the simulation on the European domain but with a different performed by changing the physics setup one-by-one through the configuration*

*Line 93 – "to investigate the optimal physics configuration for the simulation output" – it might be a bit clearer to say that the optimal physics configuration is for the accurate representation of cloud cover.*
150

We have rephrased the text as: *Large ensemble simulations with members of different physics were created in three sets to iteratively investigate the optimal configuration for cloud cover and the PV forecasting application.*

*Line 110 – What does "perform" signify here?*
155

We have rephrased the text as: *The sophisticated land surface models CLM (version 4) and Noah LSM are tested along with RUC LSM, which performs similarly to the other two LSM (Jin et al., 2010)..*

*Table 4: Why wouldn't "over-predict" in Table 4 also be a "miss"? The difference between "over" and "over-predict" isn't very clear.*

We have rephrase the text in the Section as: *Here, we use the convention that "under" and "over" represent the partial matches between fully missed or false clouds.*

*Line 131 – Can the authors define what a "rater" is here? If CFC data aren't available over Northern Europe, why wasn't cloud cover verified over the western and southern portion of the simulation domain?*

We have rephrased the text as: *The Kappa ($\kappa$) score is used to measure agreement between two or more raters, using determination in large data sets like for subjects in psychological research (Fleiss and Cohen, 1973).*

We did not discuss the cloud cover over the western and southern domain to avoid the uncertainty from the sea-atmospheric interaction. Motivated by the study on the renewable energy in Germany, our simulation use the design domain (Germany-centered) to investigate the simulation results against the observation.

*Lines 169-171 – The explanation of how/if the CFC data are upscaled for verification is unclear.*

We have added more details and rephrased the text as: *In order to compare the satellite data to the lower resolution model results, we simply average the CFC pixels within each grid cell[1]. The target value for any model grid point is then averaged over 12 to 36 observation points, depending on the location.*

*Figures 6 and 7, 9 and 10 – Why where the specific dates chosen for these figures?*

We have explained in Session 3.1 as: *These are more or less random days in different months without rare conditions, as we target the general forecasting performance for PV. Due to limited computational resources we are only able to demonstrate Sets 1 & 2 & 3 in ESIAS-met for day-ahead simulations beginning on these six days.*

*Lines247-250 – This text is unclear. Please clarify.*
* * *
[1]Other studies like (Bentley et al., 1977) may use all points within a fixed radius, which may or may not overlap

We have clarified it as: *However, the temporal- and spatial-averaged cloud covers provide less information and less vari-*
ability over time. To determine the simulation skill on the spatial patterns, we score the simulation result by calculating the
*Kappa score using the pixels in the simulation domain.*

*Lines 329-332 – It is unclear how many ensemble members exist in this study.*

We have updated the description as: *We identify the most sensitive physics clusters and use the physics configuration (Set
3) to generate totally 1,152 ensemble members. Each physics configuration simulates with 31 additional members employing
SKEBS scheme.*

*Line 344 – A deterministic simulation is never going to be unbiased.*

Agree, we have rephrased the text as: *The most accurate configuration for a deterministic forecast may differ from that for
the ensemble with the most accurate mean or that best captures the uncertainty and diversity of possible outcomes.*

*Lines 349-350 – Note that Jankov et al. (2019) don't necessarily advocate for multi-physics over stochastic-based ensem-
bles. The authors describe the practical and theoretical deficiencies of multi-physics ensembles as well.*

Firstly we appologize for mistaking the reference – We refer it to Jankov et al. (2017). Second we agree they did not ad-
vocate the multi-physics ensembles since they describe the disadvantage of multi-physics ensembles in the introduction session.

*Line 352 – Spread produced by a multi-physics-based ensemble is mostly due to physics biases, not physics uncertainty.*

Agree, and we have rephrased it as: *In our simulations, combining the ensembles into one multi-physics ensemble would
enhance the spread, but this would be somewhat artificially due to the different biases of the model physics.*

*Lines 354-355 – I wouldn't call this random. It's a specific result of the different physics parameterizations. It's also possible
that probability matched mean could be calculated for the cloud field instead of just the standard mean to alleviate some of
these problems.*

Agree, we have modified it as *biases of the model physics*.

*Line 356 – Jankov et al. (2019) used an eight-member ensemble, not four.*

We have fixed it to *yielding eight members in total*. However, in Jankov et al. (2017) it's four ensemble members for one physics configuration.

225

*Abstract – the following two sentences aren't clear: "We then selectively conduct stochastic simulations to assess the best choice for ensemble forecasts. The results indicate a high variability in terms of physics and parameterization."*

We have rephrased the abstract and changed it to *on six test cases*.

230

*Line 23 – "negative wind energy prices" – This topic needs to be briefly explained*

We explained it by rephrasing it as: *e.g. in negative prices during high wind events*.

235   *Line 23 - to study to the -> "to study the"*

The sentence is rephrased and thus this error is not existed

*Line 26-29 - The introduction to deterministic models should be followed by a reference to the WRF model as being deter-*
240   *ministic. Something like "Various global and regional deterministic weather models"*

We have changed to use *Various global and regional deterministic weather models*.

*Line 31 – "optimal meteorological models," – I would say "optimal model configuration" instead*

245

We have changed the text to *optimal model configuration*.

*Lines 39-41 – Double check for typos and correct comma placement.*

250   The typos and comma are double-checked.

*Table 4 – Typo in description -> "Indaddition"*

We have fixed the typo.

255

*Line 206 – Typo - "most" -> "the most"*

We have fixed the typo.

**References**

User's Guide for the Advanced Research WRF (ARW) Modeling System Version 3.7, https://www2.mmm.ucar.edu/wrf/users/docs/user_guide_V3/user_guide_V3.7/users_guide_chap5.htm, 2015.

Bentley, J. L., Stanat, D. F., and Williams, E.: The complexity of finding fixed-radius near neighbors, Information Processing Letters, 6, 209–212, https://doi.org/https://doi.org/10.1016/0020-0190(77)90070-9, https://www.sciencedirect.com/science/article/pii/0020019077900709, 1977.

Berndt, J.: On the predictability of exceptional error events in wind power forecasting–an ultra large ensemble approach–, 2018.

Fleiss, J. L. and Cohen, J.: The equivalence of weighted kappa and the intraclass correlation coefficient as measures of reliability, Educational and psychological measurement, 33, 613–619, 1973.

Franke, P., Lange, A. C., and Elbern, H.: Particle-filter-based volcanic ash emission inversion applied to a hypothetical sub-Plinian Eyjafjalla-jökull eruption using the Ensemble for Stochastic Integration of Atmospheric Simulations (ESIAS-chem) version 1.0, Geoscientific Model Development, 15, 1037–1060, https://doi.org/10.5194/gmd-15-1037-2022, https://gmd.copernicus.org/articles/15/1037/2022/, 2022.

Jankov, I., Berner, J., Beck, J., Jiang, H., Olson, J. B., Grell, G., Smirnova, T. G., Benjamin, S. G., and Brown, J. M.: A Performance Comparison between Multiphysics and Stochastic Approaches within a North American RAP Ensemble, Monthly Weather Review, 145, 1161–1179, https://doi.org/10.1175/MWR-D-16-0160.1, https://journals.ametsoc.org/view/journals/mwre/145/4/mwr-d-16-0160.1.xml, 2017.

Jin, J., Miller, N. L., and Schlegel, N.: Sensitivity study of four land surface schemes in the WRF model, Advances in Meteorology, 2010, 2010.

---

## Author Comment (AC2)

**Authors' response to reviewer 2: Optimization of weather forecasting for cloud cover over the European domain using the meteorological component of the Ensemble for Stochastic Integration of Atmospheric Simulations version 1.0**

Yen-Sen Lu[1,4], Garrett Good[2], and Hendrik Elbern[3]

[1]Institute of Energy and Climate Research – Troposphere (IEK-8), Forschungszentrum Jülich GmbH, 52425 Jülich, Germany
[2]Fraunhofer Institute for Energy Economics and Energy System Technology IEE, Königstor 59, 34119 Kassel, Germany
[3]Rhenish Institute for Environmental Research at the University of Cologne, Cologne, Germany
[4]Jülich Supercomputing Centre, Forschungszentrum Jülich, Jülich, Germany

**Correspondence:** Yen-Sen Lu (ye.lu@fz-juelich.de)

First of all, we thanks reviewer 2 for giving us a nice and detailed review to improve our manuscript. We read the comments and will reply to the following comments.

**1   Reply to the *General Comments**

5   *My main concern is that there is no justification that the 6 cases provide enough information about the variety of cases that these parameterizations experience when in an operational model. Are the case characteristics representative of the variability in weather patterns across the domain? Does this collection of 6 cases contain passing fronts, extreme weather, and calm conditions? Why are there no winter cases included?*

10   The reviewer is right to point out the compromise made in the number of physics simulated versus their representativeness due to computational expense. We discuss this in the revised manuscript and have been able to supplement the results with a long-term (6 month) simulation in now section 4.5 to help test the operational performance of the most promising physics configurations.

15   We have also tried to add clarity to the framing and objective of the paper regarding the (1) iterative approach of starting with a limited study of a very large number of physics and narrowing this down to fewer physics studied with more detailed simulations, and (2) the objective of this research, which is to optimize WRF specifically for cloud cover and the solar power application, which was the goal of the funding project.

20   *Why only examine cloud cover? Simply using the fraction of a column covered by cloud could obscure important model de-ficiencies like putting the clouds too high, for example. Surely, the amount of light reaching the surface is different if the cloud cover comes in the form of cirrus instead of boundary layer clouds. The general conclusions of this work could be altered if, for example, column aerosol optical depth were considered instead of cloud cover. Colum AOD is crucial for modeling pollution transport and boundary layer physics packages might play a more significant role (of course scavenging in the microphysics*

25 *parameterization will also be important).*

We hope the motivation is clearer in the new text that we focus on clouds, as the aim of this research is to improve the choice of physics for the operational accuracy of WRF for solar power in Europe, without the complexity of assessing the specific deficencies or types of clouds of the many physics, though these are all good points for future research. The choice of cloud

30   fraction in particular is also determined by the satellite observation data product.

The effect of aerosol on the cloud formation is very important and may change a lot. We'd planned to fully couple ESIAS-met and ESIAS-chem, i.e. WRF3.7.1 and EURAD-IM, to study the feedback between aerosols and cloud, but this task requires proper funding to complete.

35

*The authors do not utilize a satellite simulator package in order to make fair comparisons between models and observations. I am concerned that model is looking "straight down" at each column's respective zenith when computing cloud cover but the SEVERI instrument is observing at a sharp angle (some observations are made above the arctic circle from geostationary orbit!). The lack of cloud height information could potentially lead to mis-placed clouds. Have the authors noticed any persis-*

40 *tent biases or noise related to the zenith angle of the SEVERI observations? In addition to cloud height-related issues, every observation comes with a minimum detectable signal but models mostly do not. For example, truly-existing thin cirrus may be undetected by SEVERI due to weaknesses in infrared detection of clouds and algorithm deficiencies. A satellite simulator would alleviate these issues a great deal, if implemented correctly.*

45   Due to the moderate resolution of 20km in our comparison and the use of a discrete cloud mask, we calculate that this effect is negligible for this study, though the use of a satellite simulator is a very good point for future, higher resolution studies. We have added a paragraph on this point to section 3.2.

Due to the viewing zenith angle, the satellite observations can indeed be offset by up to a few pixels for high clouds. As

50   we aggregate the data to the 20 km grid, the inaccuracy for a high cloud could be up to one model grid point. If we however calculate how often the cloud mask has different values than its vertically neighboring grid point(s), only a few percent of the overall mask can be wrong/shifted. In the comparison to the simulation model cloud mask, where the clouds are indeed smoother than the 20km resolution, any effect on the matching rates is negligible.

55     The cloud product from CMSAF is corrected according to the methodology and algorithm by Stöckli et al. (2019), including and validated with level-2 data for instantaneous (hourly) data (Stöckli et al., 2017). For level-2 validation the difference between CFC and SYNOP data is -0.3 with a bias corrected RMSE around 30%. The application of the CFC production does not required spatial correction as well (Bojanowski and Musiał, 2020). We therefore believe this data to be suitable for our cloud mask comparison without additional calibration.

60

*There are many small typographical errors, mostly related to plurals. I noted many in the "technical corrections" but I am confident I did not document them all.*

    Thank you for indicate the typo errors. We also check it during our correction for the manuscript.

65

**2   Reply to the *Specific Comments**

*16: Which recent events in 2021?*

    We have added more specifics about recent events to the revision.

70

*95:96: "It is recommended that the surface layer physics be set with planetary boundary layer physics in WRF." Who is recommending this? Are you recommending it or is it the official recommendation from the WRF developers? It would be best if you would provide a source for this recommendation.*

75     We've updated the recommendation from the developers in the users manual as: *We note that the official documentation recommends to set the surface layer physics with specific planetary boundary layer physics in WRF (WRF, 2015).*

*102: I recognize the need for shortening the parameterization acronyms. However, these shortened acronyms are used throughout the paper and are important to the interpretation of most figures so Table A1 and Table A2 should be added to* 80 *Table 1 and Table 2. Table 1 and Table 2 have plenty of space for the shortened acronyms in parentheses behind the full names, for example.*

    Thank you for the recommendation, but we did indeed have trouble fitting it in Table 2 keep the acronyms and the physics configuration tables separate. We now merge the acronyms with the physics and parameterization tables (Table 1 and Table 2) 85 and update the text in the section 2.2 as: *The Set 1 combinations and the acronyms for the WRF physics and parameterizations are listed in Table 1.*

[Figure]

**Figure 1.** Sensitivity of threshold to matching rate for converting cloud fraction into cloud mask in the case 2015-08-23, which has the most dynamic of cloud information than other cases. The color of dot represents the difference of increase/decrease rate by applying different threshold. The base threshold is 5% and 95% for clear sky and full cloud cover, partial cloudy is in between.

*123: Are your results sensitive to these near-arbitrary thresholds?*

For the relative comparison of the model performances, we think the results are not dependent on this choice. The quantitative matching rates themselves are sensitive for some microphysics. As shown in Figure 1, the cloud mask is more sensitive to the microphysics of WSM3, WSM5, WSM6, Goddard, and CAM5.1, which can produce more cloud fraction between 0% and 100% than from other microphysics. The matching rate can increase up to 26.4%, which is produced by CAM5.1. However, with the increase of matching rate, the rate of miss (Figure 2 (a) ) and over-predict (Figure 2 (b) ) increase as well. The histogram (bin as 1%) shows that most of the cloud fraction is below 1% and overnon 100% and thus we use 5% and 95% to represent the threshold for clear sky and cloud cover, respectively.

*124: The ASOS acronym needs a definition.*

We've added the definition to the text as *Automated Surface Observing Systems*.

[Figure]

**Figure 2.** Sensitivity of threshold to the (a) missing rate and (b) over-predict rate for converting cloud fraction into cloud mask in the case 2015-08-23, which has the most dynamic of cloud information than other cases. The color of dot represents the difference of increase/decrease rate by applying different threshold. The base threshold is 5% and 95% for clear sky and full cloud cover, partial cloudy is in between.

*135: The first model evaluation results utilize this Kappa score, but there is essentially no preview of what a low-Kappa or low-Kappa means in terms of agreement with observations. Please provide some interpretation of this metric.*

We add the following text to the revision: *kappa has a maximum value of one 1 and can also be negative. The maximum kappa means a full match between two datasets. kappa between 0 and 1 indicates a partial match between the two datasets, while negative kappa indicates some anti-correlation in the matching (Pontius, 2001). A good model result should result in positive $\kappa$.*

*135-140: What is N? Total number of subjects? I am also unsure what the "subjects" are. Please provide a definition for each variable in the equations.*

We've clarified the text as : *where $\bar{P}$ is the sum of $P_i$, the matching rate of the $i^{th}$ subjects or individuals being rated, for $k$ categories. $\bar{P}_e$ is the sum of the category rate $p_j$ over $j$ and N is the total number of subjects.*

*156: Why are these cases chosen? Does the domain experience considerable variability in these cases? Fronts with strong precipitation? Mesoscale convective systems? It is very important to explain why these days were chosen so please provide a short description of each and, more importantly, why simulations of these 6 cases are capable of summarizing the variety of weather conditions that these parameterizations are expected to simulate in an operational environment. Lines 172-181 provide a cursory description of what cloud cover patterns through each case, but not a justification of why these cases are*

*sufficient to understand the differences between the parameterizations.*

We did not have the resources to simulate all physics for a wide variety of weather conditions or to target outliers, but rather begin by taking 6 somewhat random or typical days from different months. We have supplemented the results with a long-term simulation in Set 4 of six months to include diverse conditions to confirm the general performance. The results are of course not absolutely definitive, but we take this as an economical approach to a very computationally expensive problem.

*Section 3.2: Please elaborate on the description of the observational dataset. What instrument makes the observations? What techniques to they use in their cloud retrievals (BT-contrast, CO2 slicing, etc.)? What sort of processing takes the product from pixel-level to gridded, quality-controlled distribution?*

We have updated the text as: *The data is corrected and generated from SEVIRI on METEOSAT-8, which uses the visable, near infrared, and infrared wavelengths to retrieve cloud information. The hourly CFC data has level 2 validation (Stöckli et al., 2017) for the accuracy of total synoptic cloud cover and the data is corrected by the algorithm from Stöckli et al. (2019) using the clear-sky background and diurnal cycle models for brightness temperature and reflectance.*

*Figure 3: These are UTC times, right? Please state in the caption.*

The caption is updated with "UTC".

*Figure 3 caption: The caption says the colors represent both cloud cover and time of day. I think the second sentence should be removed.*

We've removed this.

*Figure 4 and Figure 5: These wallclock times would be more accessible to the reader if presented as hours, as is done with the Simulation Time. It would also be more convenient for the reader if the (a) and (b) plots had identical y-axis limits. They are very close now so why not make them identical?*

We have updated the y-axis, but use seconds as unit for counting the time, as in computational matters seconds better represent the wallclock time instead of using hours. The simulation time is counted as hours because we record the wallclock time based on the hourly timestep.

*Figure 4 caption and Figure 5 caption: There is no hourly simulation time, only total accumulated wallclock time*

We have updated the caption to match the plots (the hourly plot was removed).

*216-219: This mini-paragraph should be placed earlier in the manuscript because some science results have already been presented (Figure 6 and Figure 7). Near the first sentence in Section 4.2 or earlier would be good.*

We have moved this mini paragraph to the beginning of Section 4.2.

*255: "Accounting for the support of the simulation of the graupel mixing ratio for ESIAS-chem, we predominantly use the microphysics of WSM5, WSM6, and Goddard." is more understandable when written similar to, "We continue with the WSM5, WSM6, and Goddard microphysics parameterizations because they include treatments of graupel mixing ratio for ESIAS-chem.", unless I am misunderstanding the meaning of this sentence.*

You have captured the meaning for sure. We have updated the text as: *Accounting for the support of the simulation of the graupel mixing ratio for ESIAS-chem, we predominantly use the microphysics of WSM5, WSM6, and Goddard.*.

*281: I'm confused about the "maximum of the boxplot". In Figure 14a, the boxplot endpoints do not appear to be 1.5*interquartile range greater than the third quartile (assuming you meant quartile instead of quantile). For example, the maximum boxplot edge for the W6-T combination is only a small amount greater than the third quartile.*

Yes it's quartile and we have updated the text as: *(as maximum or minimum of the data, or 3rd quartile$\pm1.5\times$interquartile range)*. The top of the box is the 3rd quartile, and the maximum (the whiskers) of data is *limited* by the interquartile range (IQR). And therefore when the maximum/minimum of data does not greater or lower than the $1.5 \times IQR$, respectively, the whiskers does not have the length as $1.5 \times IQR$

*284: Which cumulus parameterization can improve Kappa? Tiedke?*

We have updated the text as: *Grell-3D and Tiedtke, which are more advanced than the Kain-Fritsch, can improve the Kappa.*

*Figure 14 and Figure 15 do not really add much to the analyses because they present more data than can be reasonably interpreted by the reader. The most important data are the summary data, which are only shown as text. Also, these are only two of the days and there are no analyses that summarize the other four cases! I recommend banishing the time series plots to the supplemental material and replacing these two figures with heat-maps of RMSE, sigma_bar, and x_bar and span all 6 cases.*

We believe the figures you mentioned are Figure 15 and Figure 16. We agree that the number are important to indicate the results and analysis. However, with time series we can see the comparison of spatial-average simulated cloud cover to the

[Figure]

**Figure 3.** Heat map of the RMSE between the ensemble mean total cloud fraction and observation, averaged over the last 36 hours of the simulations, shown for each cluster of microphysics and PBL physics (columns) for all test cases (rows). In the colorscale, red represents higher RMSE and poorer performance.

[Figure]

**Figure 4.** Heat map of the ensemble standard deviation ($\bar{\sigma}$) of the mean total cloud fraction, averaged over the last 36 hours of the simulations, shown for each cluster of microphysics and PBL physics (columns) for all test cases (rows). In this heat map, blue indicates larger $\bar{\sigma}$ and greater ensemble spread.

observation that changes with time. We remove Figure 16 and update Figure 15 and add the heatmap of $\bar{\sigma}$ and $\bar{x}$ to show all the results as following figures:

Also we've updated the text as: *Figure 3 summarizes the rmse performance of the ensemble mean against the observed*
195 *domain total cloud fraction. Figure 4 illustrates the time-averaged ensemble spread with the standard deviation (std) of the domain total cloud fraction. From the rmse, SBU–Lin had the best mean total cloud fraction over all six cases, with a more variable cloud fraction according to the std. The WSM3, WSM6, Goddard, and SBU–Lin with MYNN3 and ACM2 produced more accurate average cloud fractions than the other combinations. Overall, the WSM series and SBU-Lin better represented the uncertainty than the other microphysics, while MYNN3 and ACM2 improved the simulation accuracy.*

200

*310: Note that you've only investigated days during warm periods (mid-April to mid-September) so you cannot say with confidence that this is true for all time periods.*

We hope this is addressed with the revisions above.

205

*352:353: "but we should also consider the accuracy of the model physics". Yes, good point and one that is often not considered in ensemble modeling*

Thanks for appreciating this point.

210

*354:355: I think this sentence is important but I am struggling to fully understand it. Please consider rewording for better clarity.*

215 We have removed this sentence, as it does not address our point of view concisely. We have rephrased it as: *In our simulations, combining the ensembles into one multi-physics ensemble would enhance the spread, but this would be somewhat artificially due to the different biases of the model physics. The accuracies of the different model physics must then always been considered for multi-physics ensembles. We also note that the small ensemble spread reported in Jankov et al. (2017) may be due to the small number of ensembles, four for each physics configuration, yielding eight members in total.*

220 **3   Reply to *Technical Corrections***

The sentence is rephrased and thus this error is not existed.

225

The wrong Parentheses is corrected.

*Table 4 caption: Fix "Inaddition" to include a space*

230
The missing space is inserted.

235 This figure has been updated.

The sentence is rephrased and thus this error is not existed.

*216: "model is run" instead of "model is runs"*

The misspelling has been corrected.

*220: Please mention figure 8a instead of just figure 8*

This has been corrected as "Figure 8a"

*Figure 8: Please label the subplots in the figure.*

The labels have been updated.

*243: "wellby" should be "well by"*

We have correctd this misspelling.

*245: The word parameterizations is misspelled*

We have correctd this misspelling.

*265-266: You probably just want either "overall" or "over all" in this sentence and not both.*

This has been corrected in the proofreading.

*268: Missing space between sentences*

We have updated the text to include the missing space.

*Figure 15: Y-axes all say "Cloud Refraction" but they should say "Cloud Fraction"*

This figure is updated and Figure 16 is removed according to the comment above.

*308: There is a separate Conclusions section so there is no need to have "and conclusion" in this section heading.*

Agreed, *and conclusion* has been removed.

275

*339: Should say "not only changes the development of cloud cover fraction but also affects the"*

Since the PBL physics is referring to the general term of "Physics", we use the singular verb as well.

**References**

User's Guide for the Advanced Research WRF (ARW) Modeling System Version 3.7, https://www2.mmm.ucar.edu/wrf/users/docs/user_guide_V3/user_guide_V3.7/users_guide_chap5.htm, 2015.

Bojanowski, J. S. and Musiał, J. P.: Dissecting effects of orbital drift of polar-orbiting satellites on accuracy and trends of climate data records of cloud fractional cover, Atmospheric Measurement Techniques, 13, 6771–6788, https://doi.org/10.5194/amt-13-6771-2020, https://amt.copernicus.org/articles/13/6771/2020/, 2020.

Jankov, I., Berner, J., Beck, J., Jiang, H., Olson, J. B., Grell, G., Smirnova, T. G., Benjamin, S. G., and Brown, J. M.: A Performance Comparison between Multiphysics and Stochastic Approaches within a North American RAP Ensemble, Monthly Weather Review, 145, 1161–1179, https://doi.org/10.1175/MWR-D-16-0160.1, https://journals.ametsoc.org/view/journals/mwre/145/4/mwr-d-16-0160.1.xml, 2017.

Pontius, R.: Quantification error versus location error in comparison of categorical maps (vol 66, pg 1011, 2000), Photogrammetric Engineering and Remote Sensing, 67, 540–540, 2001.

Stöckli, R., Duguay–Tetzlaff, A., Bojanowski, J., Hollmann, R., Fuchs, P., and Werscheck, M.: CM SAF ClOud Fractional Cover dataset from METeosat First and Second Generation - Edition 1 (COMET Ed. 1), https://doi.org/10.5676/EUM_SAF_CM/CFC_METEOSAT/V001, https://doi.org/10.5676/EUM_SAF_CM/CFC_METEOSAT/V001, 2017.

Stöckli, R., Bojanowski, J. S., John, V. O., Duguay-Tetzlaff, A., Bourgeois, Q., Schulz, J., and Hollmann, R.: Cloud Detection with Historical Geostationary Satellite Sensors for Climate Applications, Remote Sensing, 11, https://doi.org/10.3390/rs11091052, https://www.mdpi.com/2072-4292/11/9/1052, 2019.

---

## Referee Report (RR1)

I thank the authors for responding to and incorporating reviewer suggestions while revising the manuscript. Clarity has been improved and the manuscript reads very well. I have only one major comment and minor comments that should be addressed; otherwise, this manuscript is in good shape and I would recommend it for publication after a minor revision.

Major Comments

Line 98 – The authors state that they could not evaluate Eta or Morrison microphysics due to a requirement of higher resolution; however, microphysics parameterization is necessary at all NWP resolutions, and these schemes can certainly be run at coarser scales than convection-allowing resolutions (for instance, the Eta scheme is run in the NAM model). Note that the WRF documentation states that different Eta microphysics settings should be chosen based on resolution, not that the Eta scheme shouldn't be used at all for specific resolutions:

*"e. Eta microphysics: The operational microphysics in NCEP models. A simple efficient scheme with diagnostic mixed-phase processes. For fine resolutions (< 5 km) use options (5) and for coarse resolutions use option (95)."*

Please consider revising this text.

Minor Comments

Line 39 – I would define what specific physics combination results in biased predictions of insolation over Germany, as this is most likely a physics-related problem

Line 63 – Please define a value or range for "moderate resolution"

Line 96 – A reference to Figure 2 here would be helpful.

Line 97 – "uneven spacing near the boundary layer" – Are the first eleven vertical layers more concentrated near the surface?

Line 100 – The reference to the Stergiou et al. (2017) paper may be better suited for the results section where it can be compared to the findings from this manuscript.

Line 189 – "applied to the simulation" – used as initial conditions?

Figure 5 – The caption should say "different PBL configuration"

Line 276 – Ukrain -> Ukraine

Figure 11 (and others) – Units appear to be missing for some axis titles

Line 306 – …the Goddard scheme works best…?

Line 309 – "different microphysics" – it would be good to specify which.

Figure 18 – X-axis title should by "Month of Year"

Line 407 – "insufficiencient"

Line 424 – "…comprehensive insights into **other** spatial scales, **other** meteorological variables, further physics configurations…"

---

## Author Response (AR2)

**Authors' response to reviewers: Optimization of weather forecasting for cloud cover over the European domain using the meteorological component of the Ensemble for Stochastic Integration of Atmospheric Simulations version 1.0**

Yen-Sen Lu[1,4], Garrett Good[2], and Hendrik Elbern[3]

[1]Institute of Energy and Climate Research – Troposphere (IEK-8), Forschungszentrum Jülich GmbH, 52425 Jülich, Germany
[2]Fraunhofer Institute for Energy Economics and Energy System Technology IEE, Königstor 59, 34119 Kassel, Germany
[3]Rhenish Institute for Environmental Research at the University of Cologne, Cologne, Germany
[4]Jülich Supercomputing Centre, Forschungszentrum Jülich, Jülich, Germany

**Correspondence:** Yen-Sen Lu (ye.lu@fz-juelich.de)

First of all, we authors thank the more comments from both reviewers for improving this manuscript. We also thank Editor Prof. Ullrich for editing our manuscript. All the questions and comments are replied to point by point and we also improved the manuscript over these points.

**1  Reply to the first reviewer's comments**

**1.1  Reply to the *Major Comments**

*Line 98 – The authors state that they could not evaluate Eta or Morrison microphysics due to a requirement of higher resolution; however, microphysics parameterization is necessary at all NWP resolutions, and these schemes can certainly be run at coarser scales than convection allowing resolutions (for instance, the Eta scheme is run in the NAM model). Note that the WRF documentation states that different Eta microphysics settings should be chosen based on resolution, not that the Eta scheme shouldn't be used at all for specific resolutions: "e. Eta microphysics: The operational microphysics in NCEP models. A simple efficient scheme with diagnostic mixed-phase processes. For fine resolutions (< 5 km) use options (5) and for coarse resolutions use option (95)." Please consider revising this text.*

We agree that this is not because those two microphysics can not perform under this setup. In our simulations, we did not use the correct option (95) for the Eta microphysics. And therefore we do not evaluate these simulation results over the original 674 combinations. We change the sentence to: *We thus evaluate the ten microphysics which are limited to no cloud-resolving simulations (UCAR, 2015).*

**1.2 Reply to the *Minor Comments**

*Line 39 – I would define what specific physics combination results in biased predictions of insolation over Germany, as this is most likely a physics-related problem.*

We extend the sentence to include those specific combination: *It is for example the authors' experience that WRF, using the combination of microphysics Kessler, WSM5, or WSM6 with planetary boundary layer physics YSU, MYJ, or MYNN3 Berndt (2018), typically results in biased predictions of the solar resource in Germany, which results in overestimation of solar energy in the simulation case.*

*Line 63 – Please define a value or range for "moderate resolution"*

We modify it as *(at moderate resolution, between convection-permitting and global high-resolution (Kitoh and Endo, 2016))*

*Line 96 – A reference to Figure 2 here would be helpful.*

We add the reference length accordingly.

*Line 97 – "uneven spacing near the boundary layer" – Are the first eleven vertical layers more concentrated near the surface?*

Yes, and thus we modify it as *uneven spacing through the vertical direction, especially near the surface*

*Line 100 – The reference to the Stergiou et al. (2017) paper may be better suited for the results section where it can be compared to the findings from this manuscript.*

We extend the discussion with some comparison to Stergiou et al. (2017) at line 435: *Our results show some agreement with Stergiou et al. (2017), that WSM6 and Goddard was performing well (as TOPSIS Ranking as 5th and 7th) for the precipitation in July and CAM5.1 performs good for the temperature in January.*

*Line 189 – "applied to the simulation" – used as initial conditions?*

We modify our sentence to: *is also used as initial conditions*

*Figure 5 – The caption should say "different PBL configuration"*

Yes we fix this typo from *microphysics* to *PBL*.

*Line 276 – Ukrain -> Ukraine*

We fix this typo.

*Figure 11 (and others) – Units appear to be missing for some axis titles*

For Kernel Density and cloud fraction it's unitless, which is marked as (-)

*Line 306 – . . . the Goddard scheme works best. . . ?*

Yes. we modify it to: *By row, the Goddard works better with the Tiedtke and Grell-3D cumulus ... .*

*Line 309 – "different microphysics" – it would be good to specify which.*

We modify the sentence as: *perform better with WSM5, WSM6, and Goddard microphysics*

*Figure 18 – X-axis title should by "Month of Year"*

We fix this typo.

*Line 407 – "insufficiencient"*

We fix this typo to: *insufficient*

*Line 424 – "...comprehensive insights into other spatial scales, other meteorological variables, further physics configurations..."*

55    We fix this to ... *comprehensive insights into other spatial scales* ...

**2    Reply to the second reviewer's comments**

**2.1    Reply to the *General Comments**

*My only general comment regards the new and final figure, which for some reason plots individual values of the matching rates instead of utilizing the kappa score, which was used in all of the other analyses. As it is now, Figure 18 is not very useful*
60    *because there are just too many points for the eye to easily interpret. With these longer simulations, the authors should consider compositing the hourly matching rates based on environmental factors (local time of day?, week of the year?, synoptic setting?, etc.) and present the kappa values computed in each of the bins. Doing so would provide a much more convincing argument that the simulations with these four models are similar.*

We change the original dot plot into the boxplot that shows the variation within a month for better interpretation. Also we
65    use change to use the kappa to show a more comprehensive result over the comparison between simulated and observed cloud mask.

**2.2    Reply to the *Specific Comments**

*138: How are they biased?*

We believe you asked the same question with the first reviewer for Line 39, and thus we've already extended this sentence
70    to clarify.

*Eq 3: I think this should be pi instead of pj because the outer-sum in Eq 2 is over i and not j and also the sum in Eq 3 is already over j so there are no remaining j-specific variables. But then in Line 160 it says "pj over j" so I am very confused. I also do not understand why the final n in Eq 2 and Eq 3 is in parentheses. It also seems that I do not understand the difference between n and nij. The description in the "2.3.2 Kappa score" section seems like a textbook description but it would be more*
75    *useful for the reader if the variables were introduced in relation to how they are used in this specific work, for example by relating them to the elements in Table 5.*

We have re-worked on the formula to make it more clear for reader. We also add the description connecting to our simulation work as: $\bar{P} - \bar{P}_e$ *is the actual degree of agreement between raters and* $1 - \bar{P}_e$ *is the degree of agreement when matching correctly. For number of* $n$ *raters,* $N$ *subjects will be rated into* $k$ *categories. Each* $n_{ij}$ *represents the number of raters agreeing*
80    *on the* $j$*-th category. In our work, the five possible outcomes in Table 5 are the categories for the two raters, the simulation and observation results for the gridcells as subjects.*

*Figure 18: Why did you choose to examine the matching rate instead of the kappa score like was done in the rest of the analyses?*

We change it to the kappa score for consistency. Using matching rate is convenient when the cloud mask data shows only two
85 data type including cloudy and clear sky. Using matching rate can be straight forward for showing only 3 categories (match, miss, and false).

**2.3  Reply to the *Technical Comments**

*160: I suggest moving Eq 2 and Eq 3 up to directly follow Eq 1 because the text on lines 159-160 includes variables that have not yet been shown in an equation.*

90 Yes, I've follow this suggestion for working on the specific comment.

*167: The variable n is introduced in a different section than where it is first used (Eq 2).*

We have re-worked on the equations, as your specific comment.

*Figure 3: Increase the font size in this and many of the other figures.*

We modify the figures accordingly, including Figure 3, 4, 5, 8, 9, 10, 11, 14, 16, and 17.

95 *242: There is no Figure 4c or Figure 5c. I think you mean Figure 4a and Figure 5a but I'm not completely sure.*

Yes, it's 4b and 5b, we fix this typo.

*Figure 5 caption: This says microphysics configuration but it should be PBL configuration, according to lines 232-233.*

Yes, we fix this typo.

**References**

100    Berndt, J.: On the predictability of exceptional error events in wind power forecasting–an ultra large ensemble approach–, Ph.D. thesis, Universität zu Köln, 2018.

Kitoh, A. and Endo, H.: Changes in precipitation extremes projected by a 20-km mesh global atmospheric model, Weather and Climate Extremes, 11, 41–52, https://doi.org/https://doi.org/10.1016/j.wace.2015.09.001, https://www.sciencedirect.com/science/article/pii/S2212094715300219, observed and Projected (Longer-term) Changes in Weather and Climate Extremes, 2016.

105    Stergiou, I., Tagaris, E., and Sotiropoulou, R.-E. P.: Sensitivity Assessment of WRF Parameterizations over Europe, Proceedings, 1, https://doi.org/10.3390/ecas2017-04138, 2017.

UCAR: User's Guide for the Advanced Research WRF (ARW) Modeling System Version 3.7, https://www2.mmm.ucar.edu/wrf/users/docs/user_guide_V3/user_guide_V3.7/users_guide_chap5.htm, 2015.